# Hydrothermal Synthesis of Sodalite-Type N-A-S-H from Fly Ash to Remove Ammonium and Phosphorus from Water

**DOI:** 10.3390/ma14112741

**Published:** 2021-05-22

**Authors:** Pengcheng Lv, Ruihong Meng, Zhongyang Mao, Min Deng

**Affiliations:** 1College of Materials Science and Engineering, Nanjing Tech University, Nanjing 211800, China; 201861203143@njtech.edu.cn (P.L.); mzy@njtech.edu.cn (Z.M.); 2State Key Laboratory of Materials-Oriented Chemical Engineering, Nanjing Tech University, Nanjing 211800, China; 3Guodian New Energy Technology Research Institute Co., Ltd., Beijing 102209, China; mrh0210@163.com

**Keywords:** N-A-S-H, sodalite, simultaneous removal, ammonium, phosphorus

## Abstract

In this study, the hydrated sodium aluminosilicate material was synthesized by one-step hydrothermal alkaline desilication using fly ash (FA) as raw material. The synthesized materials were characterized by XRD, XRF, FT-IR and SEM. The characterization results showed that the alkali-soluble desilication successfully had synthesized the sodium aluminosilicate crystalline (N-A-S-H) phase of sodalite-type (SOD), and the modified material had good ionic affinity and adsorption capacity. In order to figure out the suitability of SOD as an adsorbent for the removal of ammonium and phosphorus from wastewater, the effects of material dosing, contact time, ambient pH and initial solute concentration on the simultaneous removal of ammonium and phosphorus are investigated by intermittent adsorption tests. Under the optimal adsorption conditions, the removal rate of ammonium was 73.3%, the removal rate of phosphate was 85.8% and the unit adsorption capacity reached 9.15 mg/L and 2.14 mg/L, respectively. Adsorption kinetic studies showed that the adsorption of ammonium and phosphorus by SOD was consistent with a quasi-secondary kinetic model. The adsorption isotherm analysis showed that the equilibrium data were in good agreement with the Langmuir and Freundlich model. According to thermodynamic calculations, the adsorption of ammonium and phosphorus was found to be a heat-absorbing and spontaneous process. Therefore, the preparation of SOD by modified FA has good adsorption properties as adsorbent and has excellent potential for application in the removal of contaminants from wastewater.

## 1. Introduction

With the development of urbanization, eutrophication of water bodies is a growing problem due to the excessive direct or indirect discharge of ammonium and phosphorus [1,2]. Ammonia nitrogen and phosphorus are present in the water bodies as ammonium ions and phosphates, and if the ammonium ion and phosphate content exceed the standard this will lead to water quality deterioration and toxic effects [3]. Eutrophication leads to the excessive bloom of planktonic algae that disrupts the ecological balance of water bodies [4]. Currently, there are a wide variety of methods to remove phosphate and ammonium, wherein chemical precipitation techniques, biological treatment methods, ion-exchange methods and adsorption methods were roughly included [5,6,7]. Ammonium ions were not easily precipitated and usually removed by ion exchange during the adsorption process [8]. Phosphate removal was achieved by the formation of insoluble phosphate precipitates with metal ions [9]. The results showed that direct adsorption of ammonium and phosphate with adsorbents was the optimal solution for removal [10,11]. Therefore, the simultaneous removal or recovery of multiple contaminants from wastewater has become a challenge in wastewater treatment technology. The preparation of convenient, economical and efficient adsorbent materials has also become one of the hot research topics in recent years.

Fly ash (FA) is one of the most massive industrial solid wastes discharged in China and currently on earth [12]. Due to the superimposed influence of various factors such as the nature of raw materials and formation conditions. The resulting FA has a complex composition structure and contains trace toxic elements, which limits the sustainable resource utilization of FA [13]. In the process of exploring the high value-added utilization of FA, the chemical composition of fly ash (wt%SiO_2_ + wt%Al_2_O_3_ > 70%) was found to be extremely similar to that of zeolite [14]. The first successful derivation of zeolite materials from fly ash was made by Holler and Wirsching as early as 1985 [15]. Nowadays, highly efficient adsorbent materials prepared from loaded modified fly ash have been widely used for the removal of nutrient-rich pollutants from wastewater. Ji et al. [16] synthesized zeolites with silica-rich (SF-Z) and calcium-rich (CF-Z) FA, respectively, as raw materials, and found that they had good solidification to ammonium and phosphate in wastewater. You and Valderrama et al. [17] used FA synthesized Zeolite P loaded with calcium and magnesium ions (Ze-Ca, Ze-Mg) adsorbent materials to recover both ammonium and phosphate from simulated treated wastewater. Li et al. [18] used lanthanum modified zeolite (LMZ) to modify the sediments of Taihu Lake, China, the sorption capacity of LMZ for phosphate was estimated to be 64.1 mg/g according to the Langmuir isotherm model. However, direct synthesis of porous aluminosilicate materials from fly ash is not effective in adsorption of low concentration ammonium and phosphorus containing wastewater, a few of them can remove ammonium and phosphorus from sewage simultaneously. Therefore, reasonable modifications are necessary for the direct synthesis of porous aluminosilicate materials from fly ash to enhance the ability of ammonium and phosphorus removal [19].

Sodalite-type N-A-S-H was a typical heterogeneous crystal of the Na_2_O-Al_2_O_3_-SiO_2_-H_2_O system [20]. N-A-S-H was the reaction product of alkali treatment of bauxite ore with large content, known as ‘sodium silica slag’ in production. Due to different mineralization conditions in the industrial process, “sodium-silica slag” was very easy to precipitate and bring accumulation to the pipeline and evaporator wall to affect normal industrial production [21]. If the waste slag is discharged directly, it will not only cause a lot of waste of resources of alumina and sodium oxide, but also cause secondary pollution. Thus, how to take advantage of the unique nature of the heterogeneous crystalline phase to achieve resource utilization of N-A-S-H crystals is of great significance to the development of the alumina industry and to solve the waste of resources.

In this study, a desilication process is proposed based on the preparation of zeolite-like structures by the conventional hydrothermal alkaline solution of FA. FA experiences dissolution and desilication as a single raw material under alkaline solution activation to form crystalline SOD-type N-A-S-H minerals with microporous cavities in their structure. In the experiments, the original and modified samples were systematically characterized by XRD, XRF, FTIR, SEM, etc. The modified products were also used as adsorbents to investigate the simultaneous removal of nitrogen and phosphorus from wastewater by intermittent adsorption. The study explored the influence of adsorbent dosage, contact time and environmental factors on the adsorption effect. Common adsorption models such as Lagergren first-order, Ho’pseudo-second-order, Langmuir isotherm model, Freundlich isotherm model, etc. were selected to describe the adsorption process, and other possible removal mechanisms were considered.

## 2. Materials and Methods

### 2.1. Materials

The raw material used in this experiment was FA from Huaneng Jinling coal-fired power plant in Nanjing, China and its chemical composition was analyzed in Table 1. Analytically reagent (AR ≥ 99.5% purity) anhydrous KH_2_PO_4_ and guaranteed reagent (GR ≥ 99.9% purity) NH_4_Cl were supplied by Sinopharm Chemical Reagent Co., Ltd., Shanghai, China. Sodium hydroxide (AR, ≥96.0% purity) was obtained from Aladdin Industrial Corporation, Shanghai, China. For the experiments, deionized water was used for the configuration of all solutions.

### 2.2. Synthesis of SOD

Combined with the current methods on the comprehensive utilization of fly ash, a one-step hydrothermal method of alkali dissolution desilication with simple operation and the low energy consumption is proposed to synthesis hydrated sodium aluminosilicate [22]. Firstly, fly ash is dried at 105 °C for 24 h to constant weight, and then pre-treated by mechanical grinding and high temperature scorching. The purpose of this work was to effectively increase the contact area of the powder particles, improve the activity, remove the residual organic matter such as unburned charcoal and enhance the purity of the later synthesis products. The 4 mol/L alkalines NaOH aqueous solution was selected to leach FA, add 10 g of raw ash and dissolve silicon with a liquid-to-solid ratio of 5:1. The liquid-solid mixture was stirred at a uniform speed of 120–140 rpm/min at 80 °C for 6 h. Then accelerated vigorously and stirred at a higher temperature of 90 °C to form SOD-type N-A-S-H with very little solubility. After sufficient cooling, the synthesized product was filtered and washed with water until the pH of the upper clear layer was about 9. Finally, the samples were dried in an oven at 105 °C for 24 h to a constant weight. The synthesis product was placed directly into the dryer without grinding.

### 2.3. Characterization Techniques

The raw material FA and the synthesized SOD-type N-A-S-H were firstly sieved through an 80 μm sieve to get samples. The X-ray Powder Diffraction (XRPD) analysis was performed by using a Rigaku Smart Lab (3) X-ray diffractometer (Cu target, voltage 3 kw, XRD scan range 10–70°, step size 0.02°, scan speed:10°/min, Rigaku, Tokyo, Japan). Fourier transform infrared (FT-IR) spectra of each solid product were recorded by an FTIR 920 Fourier transform infrared spectrometer (Nexus 670, Thermo, Waltham, MA, USA). Detailed operations were shown as below: sampled in dry air, 1 mg sample was fully grinded with 200 mg KBr and pressed into sheets, and the FT-IR spectra of samples were achieved at a resolution of 2 cm^−1^ in the wave number range of 4000–400 cm^−1^. The morphologies of both samples after gold application were observed by Zeiss Ultra Plus field emission SEM (Ultra55 FE, Zeiss, Berlin, Germany). The internal pore structure and specific surface area of the materials were examined by N_2_ adsorption BET by use of a Micro-meritics ASAP 2020 volumetric instrument (Micromeritics Instrument Corporation, Norcross, GA, USA). The chemical composition of the raw ash and the synthesized products were determined by a Nihon Rei Mini-Z X-ray fluorescence spectrometer (XRF) (PANalytical, Almelo, Netherlands). The Ca^2+^ content in the adsorbed solution was measured by diluting the solution to be measured using an inductively coupled plasma emission spectrometer (ICP-OES) model ICP-5000 (PG Instruments, London, UK).

### 2.4. Batch Experiments

In this experiment, the practical effectiveness of the synthetic product SOD-type N-A-S-H for the simultaneous removal of nitrogen and phosphorus from domestic wastewater was investigated in the laboratory using a static intermittent adsorption method. All adsorption experiments were performed in 250 mL conical flasks with continuous stirring at 180 rpm/min at room temperature of 25 °C. Anhydrous KH_2_PO_4_ (AR) and NH_4_Cl (GR) were used as raw materials to prepare 1000 mg/L ammonia nitrogen standard reserve solution and 500 mg/L phosphorus standard reserve solution. In the experiment, the standard solution was diluted to the experimentally required concentration by adding deionized water, and the dilution was corrected when calculating the results. The pH of the water samples was controlled by adding preconfigured 1 mol/L HCl and NaOH solutions. The optimum adsorption conditions for simultaneous ammonium and phosphorus removal were determined by considering important factors such as adsorbent dosage (0.5–6 g/L), adsorption time (15–180 min with 15 min as one interval) and pH value of adsorption medium (3–11). Three parallel experiments were set up for each group, and the mean data were calculated with error bars indicating the standard deviation.

Based on the general ammonium and phosphate concentration ratio of N/P = 5 in Chinese domestic wastewater, the initial ammonium and phosphate concentrations are configured to be 50 mg N/L and 10 mg P/L [23]. Under the premise of determining the best experimental conditions such as the amount of adsorbent and adsorption time, the initial concentrations of ammonium nitrogen and phosphate were changed to study whether there is a competitive relationship between simultaneous adsorption of ammonium ions and phosphate. When the initial concentration of ammonium (C_N_) in the solution is 50 mg/L, the initial concentration of phosphate (C_P_) was adjusted between 10 and 100 mg/L. When C_P_ = 10 mg/L, adjusted C_N_ = 50–500 mg/L.

By analyzing the changes in adsorption capacity during solute adsorption per unit of adsorbent at different time intervals, the kinetic processes of ammonium and phosphate adsorption were studied and other kinetic models were used to fit the experimental data. The adsorption isotherm experiments were performed at three temperatures (25, 35 and 45 °C).

For all batches of experiments in the above study, the supernatant of the experimental water samples was filtered with a 0.45 mm cellulose acetate membrane for each test separation. Residual ammonium and phosphate in solution were detected by use of a spectrophotometer (UV-1800, Instrument, Shanghai, China) [24]. Ammonium concentration was determined by Alkaline potassium persulfate digestion UV spectrophotometric method [25], phosphorus concentration was determined by Ammonium molybdate spectrophotometric method [26]. The unit adsorption capacity (*Q_e_* mg/g) and removal rate of the synthetic material (%R) for ammonium and phosphate are calculated by the following equations. Equations (1) and (2).
(1)$$Q_{e}=\frac{C_{0}-C_{e}}{W}\times V$$
(2)$$Removal\;rate\;(\%)=\frac{C_{0}-C_{e}}{C_{0}}\times 100$$
wherein *C*_0_ and *C**_e_* are the initial and residual ion concentrations in the solution at time *t* (mg/L), respectively. *V* represents the volume of the solution (L) and W is the mass of the adsorbent (g).

### 2.5. Saturated Adsorbent Desorption Regeneration Experiment

A standard stock solution of ammonium ion and phosphate (C_N_ = 1000 mg/L, C_P_ = 500 mg/L) were diluted with deionized water. Dilute to the concentration of ammonium ion and phosphate, respectively, C_N_ = 500 mg/L and C_P_ = 200 mg/L. The 4 g/L of SOD was added to the two solutions, and the saturated adsorption material was prepared by stirring and adsorbing for 12 h under the best adsorption conditions (*T* = 25 °C, pH = 8) in the experiment. After the adsorption equilibrium, the supernatant solute content was determined by standing. The adsorbed saturated SOD was washed several times during filtration, then put in an oven to dry and placed in a desiccator for use.

The desorption cycle regeneration experiment was prepared by drawing on the method of adsorption regeneration of saturated zeolite materials. In view of the different adsorption mechanisms of different solutes, 1 mol/L NaCl was selected as the desorption liquid for SOD saturated with adsorbed nitrogen, and 1 mol/L CaCl_2_ was selected as the desorption liquid when the adsorbed phosphorus was saturated. The adsorbed saturated SOD was put into the desorption solution, stirred at 25 °C for 3 h and the concentration of solute in the solution was measured, and the desorbed SOD was washed and dried. The desorption rate could be calculated from the relationship between the saturated adsorption capacity and the adsorption capacity after desorption, as shown in Equation (3). The desorbed SOD was the regenerated SOD, which continued to be cast into the saturated solution for adsorption under optimal adsorption conditions. The concentration of solute in the solution after adsorption was measured to calculate the regeneration rate of SOD, which was calculated as in Equation (4).
(3)$$Desorption\;rate\;(\%)=\frac{Q_{D}}{Q_{\max}}\times 100\%$$
(4)$$\mathrm{Regeneration}\;rate\;(\%)=\frac{Q_{R}}{Q_{\max}}\times 100\%$$

In the formula, *Q_max_* represents the unit adsorption amount of saturated adsorption, the unit is mg/g. *Q_D_* is the unit adsorption amount dissolved out during the desorption process, and the unit is mg/g. *Q_R_* is the unit adsorption amount of re-adsorption after desorption, and the unit is mg/g.

## 3. Results and Discussion

### 3.1. Characterization of FA and SOD

The XRD patterns of fly ash and its synthetic N-A-S-H are shown in Figure 1. The measured XRD data were analyzed by Jade 6.0 software and matched with the mineral standard diffraction pattern to determine the mineral composition. The results show that the main crystalline phases of FA used in this study are mullite (Al_6_O_13_Si_2_) and quartz (SiO_2_), containing large amounts of silicon and aluminum components. The XRD pattern contains a clear broad between 20° and 30°, indicating the presence of an amorphous glass phase in the original ash [27]. After the fly ash is heated and de-siliconized by alkali dissolution, a small amount of quartz is dissolved with the glass phase and the aluminosilicate gel formed is transformed into the less soluble N-A-S-H. The specific reaction process is derived from the following Equations (5) and (6).
(5)$$H_{2}\mathrm{Si}O_{4}{{}^{2-}}_{\left(\mathrm{aq}\right)}+\mathrm{Al}{\left(\mathrm{OH}\right)}_{4}{{}^{-}}_{\left(\mathrm{aq}\right)}\to {\left[{\mathrm{Al}}_{2}\left[H_{2}\mathrm{Si}O_{4}\right]{\left(\mathrm{OH}\right)}_{6}\right]}^{2-}{}_{\left(\mathrm{aq}\right)}+{\mathrm{OH}}^{-}{}_{\left(\mathrm{aq}\right)}$$
(6)$${\mathrm{Na}}^{+}{}_{\left(\mathrm{aq}\right)}+{\left[{\mathrm{Al}}_{2}\left[H_{2}\mathrm{Si}O_{4}\right]{\left(\mathrm{OH}\right)}_{6}\right]}^{2-}{}_{\left(\mathrm{aq}\right)}\to {\mathrm{Na}}_{2}O\cdot {\mathrm{Al}}_{2}O_{3}\cdot n\mathrm{Si}O_{2}\cdot mH_{2}O+H_{2}O+H^{+}{}_{\left(\mathrm{aq}\right)}$$

The chemical composition of the synthesized product is similar to that of the original ash (Table 1), formed the sodium aluminosilicate with a low SiO_2_/Al_2_O_3_ ratio. Therefore, the solution has more free hydroxide ions, and the generated sodium aluminosilicate is dominated by hydroxy Sodalite [28]. Combined with the XRD pattern, it can be seen that a single SOD-type hydrated sodium aluminosilicate (Na_2_O·Al_2_O_3_·1.7SiO_2_·2.4H_2_O) is generated. The XRD peaks of the by-product mullite phase do not diminish, which confirms that mullite remains a stable mineral phase under low-temperature alkali dissolution experimental conditions [29]. After calculation of the crystallinity of the physical phase by software, it is found that the content of SOD physical phase is lower than expected. The reason is analyzed to be that the dissolution of Ca compounds by alkali form calcium silicate, as well as undissolved stable mullite, inhibits SOD synthesis [30].

The FTIR spectra of the pre- and post-modification materials are shown in Figure 2. It can be seen that the fly ash has a peak at 1090 cm^−1^, which belongs to the asymmetric stretching vibration region of T-O (T-Si or Al). The T-O asymmetric stretching vibrational region corresponding to the SOD is shifted to 991 cm^−1^ due to the incorporation of aluminum atoms into the tetrahedral framework. The internal bending vibration of the two samples T-O is between 450 and 500 cm^−1^ [31]. The continuous bands are detected by SOD at 660 and 720 cm^−1^ correspond to the bicyclic vibrations of tetrahedral Si-O and Si-O-Al, respectively. The band detected around 1647 cm^−1^ belonging to the common bending vibration of water molecules, while the sharp band appearing around 3547 cm^−1^ corresponds to the OH stretching of water molecules, which indicates the hydrophilic nature of the synthesized product.

The morphological analysis of fly ash (Figure 3a) and synthesized products (Figure 3b) by SEM presents that after transformation of the surface of the original ash with a smooth and irregularly rounded spherical structure the aluminosilicate gel is enriched on the surface to form a well-arranged and porous cylindrical petal-like crystal structure.

BET method is the abbreviation for BET specific surface area detection method. The BET method can test the specific surface area, pore volume, pore size and distribution of the material and distinguish the shape and structure of the pores according to the shape of the adsorption-desorption curve. Figure 4a is the nitrogen adsorption/desorption isotherm diagram in the BET test, the results showed that the specific surface area of the synthesized product increases from 1.58 m^2^/g of the original ash to 60.54 m^2^/g, which is 22 times bigger than the original. Figure 4b presents the BJH results of the pore size distribution which shows that there are a large number of nanoscale micropores in the synthesized product SOD. The cation exchange capacity (CEC) value increases from 7.5 cmol/kg before synthesis to 136.5 cmol/kg, and the cation exchange capacity of SOD is much larger than that of the original ash.

Combined with various characterizations, the new crystalline phase is identified as SOD-type N-A-S-H. The crystalline aluminosilicate structure of SOD consists of [SiO_4_]^4−^ and [AlO_4_]^5−^ tetrahedra through shared oxygen atoms, and the tetrahedra form a three-dimensional network with many voids and open spaces. These voids define various special properties of zeolite, such as absorbing molecules in huge internal channels. Si^4+^ will be replaced by Al^3+^ in the tetrahedra, resulting in a negative charge in the SOD structure [32]. When open space allows the entry of cations, higher CEC may be resulted. The modification has a large number of voids allowing the SOD to use the substantial internal pore structure to adsorb molecules [33]. Thus, zeolite synthesized from waste ash has potential application in wastewater treatment and has been used as an adsorbent to remove selected pollutants in wastewater.

### 3.2. Effect of Dosage

The effect of SOD dosing as a variable on the removal of ammonium and phosphate from water is explored in this study. As the amount of adsorbent increases, the change in removal rate (Figure 5a) and unit adsorption capacity (Figure 5b) was shown in Figure 5. In the figure, TN means total ammonium, TP means total phosphorus. Under a constant concentration, the removal rate of the material for ammonium gradually improves as the adsorbent dosing increases and tends to equilibrium after a certain degree, while phosphate removal rate presents a slow adsorption rate due to the high starting efficiency. When the dosage is lower than 3 g·L^−1^, the removal rates of nitrogen and phosphorus significantly increase with the dosage increasing and show a clear increasing trend. As the dosage increases from 3 g to 6 g·L^−1^, the removal rate basically keeps stabilized without a significant increase. After this, the removal rates of ammonium and phosphorus range from 69.3% to 73.0% and 78.5% to 82.0%, respectively. The adsorption capacity decrease from 11.54 to 6.11 mg/g for ammonium and from 2.62 to 1.37 mg/g for phosphate.

For the phenomenon that removal rate increases but per unit adsorption decrease with dosage increasing, it can be attributed to the rise in unit specific surface area and adsorption sites. The increase in the dosage makes more adsorption sites on the adsorbent surface, and more sites are available for substances to be adsorbed, leading to the rise in the removal rate [34]. However, since the concentration of the solution is constant, increasing the dosage after the saturation of adsorption will reduce the adsorption capacity per unit. In summary, for considering the maximum economic benefits of practical application and the secondary pollution caused by the waste of resources as a result of overdosing, the dosage of 4 g·L^−1^ is selected in this study. The removal rate and adsorption capacity of ammonium and phosphorus removal at this time are 72.2%, 9.02 mg/g and 79.9%, 2.00 mg/g, respectively.

### 3.3. Effect of Adsorption Time

It can be seen from Figure 6 that the removal rates (Figure 6a) of ammonium and phosphorus as well as the adsorption capacity per unit of adsorbent gradually (Figure 6b) increase with the increase of reaction time. The reason for this phenomenon is due to the increased utilization of adsorption sites by the adsorbent. The removal rate and the unit adsorption amount tend to be stable at about 90 min, indicating that the sufficient reaction reached the adsorption equilibrium. Moreover, the adsorption curves flatten out with the gradual increase of adsorption time, indicating a gradual decrease of the adsorption rate. The phenomenon of ammonium and phosphorus adsorption rates are faster and then slower, mainly is a result of the maximum initial concentration of the solution and highest solvent or solute content at the beginning of the adsorption reaction. Therefore diffusion force of the liquid film is relatively large, and the solute can reach the surface of the SOD quickly. At this time, the adsorption point of the SOD is also the largest to carry out the adsorption. As the solution concentration decreases and the solute content decreases over time, and the adsorption area available for saturating the adsorption sites on the material surface gradually decreases. The membrane diffusion rate decreases, and the SOD adsorption rate gradually decreases.

In addition, the SOD is positively charged in solution, the phosphate is negatively charged and there is an electrostatic attraction between the two. The electrostatic attraction is most potent at the beginning of adsorption. As the contact time increases, the adsorbent becomes less charged, the electrostatic attraction decreases [35]. The adsorption rate of the material on phosphorus is faster, reaching 75.7% in 5 min and equilibrium in 90 min. Compared with the adsorption results in 3 h, the deviation is less than 1%. The pH slowly increases from 7.00 to 8.31 after the adjustment during the reaction. That is mainly due to the ion exchange of alkali metal ions (Na^+^, Mg^2+^, Ca^2+^, etc.) in synthesized zeolite with NH_4_^+^ ions during the reaction to improve the pH.

### 3.4. Effect of pH

The pH change of the solution affects the form of ammonium and phosphorus present in the solution, which leads to a result that the environmental factor pH often has an important effect on the adsorption capacity of fly ash and modified products. Figure 7 shows the effect of initial solution pH on the adsorption capacity of ammonium and phosphorus. It is shown that as the pH of the solution increases from 3.0 to 11.0, there are different degrees of fluctuations in the efficiency of nutrient removal from the solution.

As the pH of the solution increases from 3.0 to 11.0, the removal rate of ammonium shows an increase followed by a decrease, and the best removal effect is achieved at pH = 8. Under acidic conditions, the ammonium ions in the solution mainly exist in the form of NH_4_^+^. At pH < 5, due to the high concentration of H^+^ in the solution, H^+^ will compete with NH_4_^+^ for adsorption, making the SOD less effective in removing ammonium per unit of adsorption. With increased pH, NH_4_^+^ exists as NH_3_·H_2_O when pH > 9 (Equation (7)), resulting in poor removal rates [36]. When the pH ranges between 3–11, chosen for the experiments, phosphate is mainly present in two forms, HPO_4_^2−^/H_2_PO^4−^. At pH < 7, the concentration of H_2_PO_4_^−^ is high. As the pH ranges between 7–10, it is mainly in the form of HPO_4_^2−^. At pH > 10, H_2_PO_4_^−^ is in the majority, and a small amount of PO_4_^3−^ also presents. There is an interconversion between different forms of phosphate (Equations (8)–(10)). For phosphorus removal, an “M” shape can be observed in the graph, with the best removal at pH = 8. Under either over-acidic or over-alkaline conditions, the release of Ca^2+^ and the generation of calcium phosphate precipitation are not favorable, leading to low adsorption rates at pH < 5 and pH > 9 [37]. Observation of the adsorbed precipitates reveals the production of flocs (Figure 8), and two precipitates of calcium phosphate are found in combination with XRD (Figure 9) analysis. It is further shown that phosphorus removal is mainly by generated insoluble precipitates with Ca^2+^ and settling on the surface of Fe_2_O_3_ and Al_2_O_3_ again by ligand exchange adsorption.
(7)$${\mathrm{NH}}_{4}{{}^{+}}_{\left(\mathrm{aq}\right)}+{\mathrm{OH}}^{-}{}_{\left(\mathrm{aq}\right)}⇌{\mathrm{NH}}_{3}+H_{2}O$$
(8)$${\mathrm{PO}}_{4}{{}^{3-}}_{\left(\mathrm{aq}\right)}+H_{2}O⇌H{\mathrm{PO}}_{4}{{}^{2-}}_{\left(\mathrm{aq}\right)}+{\mathrm{OH}}^{-}{}_{\left(\mathrm{aq}\right)}$$
(9)$$H{\mathrm{PO}}_{4}{{}^{2-}}_{\left(\mathrm{aq}\right)}+H_{2}O⇌H_{2}{\mathrm{PO}}_{4}{{}^{-}}_{\left(\mathrm{aq}\right)}+{\mathrm{OH}}^{-}{}_{\left(\mathrm{aq}\right)}$$
(10)$$H_{2}{\mathrm{PO}}_{4}{{}^{-}}_{\left(\mathrm{aq}\right)}+H_{2}O⇌H_{3}{\mathrm{PO}}_{4}{{}^{-}}_{\left(\mathrm{aq}\right)}+{\mathrm{OH}}^{-}{}_{\left(\mathrm{aq}\right)}$$

### 3.5. Effect of Initial Concentration

In order to investigate whether there is competition between the simultaneous adsorption of ammonium and phosphorus by SOD, the adsorption curves are observed at different concentrations (C_N_ = 50–500 mg/L, C_P_ = 10–100 mg/L). From Figure 10 we can see that when the initial concentration of ammonium (C_N_ = 50 mg/L) keeps constant and the phosphate concentration increases from 10 to 100 mg/L, the SOD for the removal of ammonium ions from the solution remained between 72.95–74.19%. The deviation of the removal rate is around 1.24%, and this phenomenon indicated that the variation of phosphate concentration do not affect the removal of ammonium ions. On the other hand, when the initial phosphate concentration is controlled to be (C_P_ = 10 mg/L) constant, the phosphate removal rate increases with the increase of ammonium ion concentration, and the adsorption rate increases from 85.3 to 98.6%. The results show that: under the same conditions, the higher the concentration of ammonium ions, the higher the phosphate adsorption efficiency.

The above results indicate that the concentration of ammonium in solution affects the adsorption of SOD on phosphate in the presence of ammonium and phosphate coexistence. Studies show that for the removal mechanism of phosphate, it is generally shown to be removed by the generation of precipitation with Ca^2+^. To investigate the interaction that exists between ammonium and phosphate, ICP-OES is used to determine the Ca^2+^ content in deionized water and solutions containing ammonium and phosphate. The adsorbed filter residue is collected to be washed and dried on the surface, and the changes of SOD chemical composition before and after adsorption are determined using XRF. In deionized water, the Ca^2+^ concentration in the solution is measured to be 7.38 mg/L, indicating the presence of Ca^2+^ release by the addition of SOD to the solution. When the initial concentration C_P_ = 10 mg/L keeps constant, the Ca^2+^ concentration increases from 18.89 at the beginning to 58.56 mg/L as the ammonium concentration increases. Combined with the changes in chemical composition, we can see from Table 2 that the most obvious change is in the content of Na and Ca. During the treatment of simulated wastewater containing ammonium and phosphorus, the alkali metal cations (Na^+^, Ca^2+^, K^+^) in the samples are continuously exchanged out by NH_4_^+^ [8]. NH_4_^+^ replaces the cation and becomes the equilibrium ionic charge in the sample structure (Equation (11)). The ion exchange activity follows the order: Na ^+^ > Ca^2+^ > K^+^, Na^+^ is exchanged first, with the increase of ammonium ion concentration, Ca^2+^ concentration gradually increases. Both became the main objects of ion exchange, and the phosphate removal rate increases with the increase of Ca^2+^ ion concentration [38].
(11)$${\left({\mathrm{Na}}^{+}+{\mathrm{Ca}}^{2+}+K^{+}\right)}_{\mathrm{SOD}}+{\mathrm{NH}}_{4}{{}^{+}}_{\mathrm{solution}}\to {\mathrm{NH}}_{4}{{}^{+}}_{\mathrm{SOD}}+{\left({\mathrm{Na}}^{+}+{\mathrm{Ca}}^{2+}+K^{+}\right)}_{\mathrm{solution}}\left(\mathrm{aq}\right)$$

#### Adsorption Kinetics

When a fluid is in contact with a porous solid, one or more components of the fluid accumulate at the surface of the solid, which is known as adsorption. The changes in the liquid-solid adsorption process are generally expressed by a kinetic model, through which not only the reaction mechanism can be inferred, but also the solute uptake rate can be estimated. To figure out the adsorption process of ammonium and phosphate on the synthesis products, Lagergren first-order and Ho’pseudo-second-order are chosen to fit the experimental data in this study. The following Equations (12) and (13) describe the Lagergren first-order, Ho’pseudo-second-order kinetic models, respectively.
(12)$$\ln(Q_{e}-Q_{t})=\ln Q_{e}-K_{1}t$$
(13)$$\frac{t}{Q_{t}}=\frac{1}{K_{2}Q_{e}{}^{2}}+\frac{t}{Q_{e}}$$
wherein: *Q_e_*, *Q_t_* are the adsorption amounts at adsorption equilibrium and after *t* time, mg/g; *t* is the adsorption time, min; *K*_1_ (min^−1^) and *K*_2_ (g mg^−1^·min^−1^) are the adsorption rate constants for the first- and second-order models.

Figure 10 shows the fitted curves of Lagergren’s first-order model (Figure 11a) and Ho’s pseudo-second-order model (Figure 11b). The detailed parameters of the kinetic model fitting are compared in Table 3. The results showed that for ammonia nitrogen, the adsorption kinetic data all fit well the first-order model of Lagergren and the pseudo-second-order model of Ho (R^2^ > 0.9). From the higher R^2^ values, Ho’s pseudo-second-order model describes the adsorption behavior of the material for ammonium better (R^2^ > 0.99). On the other hand, for phosphate, the experimentally measured coefficient of Ho’s pseudo-second-order model (R^2^ = 0.9993) is higher than that of the Lagergren first-order model (R^2^ = 0.8107). Thus, the kinetic process of ammonium and phosphate adsorption of materials can be better interpreted by Ho’s pseudo-second-order model. The equilibrium adsorption quantities (Q_e,2_ = 9.3832, 2.1463) obtained by fitting the secondary kinetic equations are in better agreement with the actual experimental measured values (Q_e,2_ = 9.15, 2.14).

### 3.6. Adsorption Isotherm

Adsorption isotherms provide a macroscopic overview of the properties of adsorption phenomena such as adsorption volume, adsorption strength and adsorption state. It is essential to reveal the adsorption mechanism of solutes in solution by solid-phase adsorbents. Different isotherm models are characterized by different constants. They reveal the surface properties of the adsorbent and the affinity between the adsorbent and the adsorbed material. For ammonium and phosphate adsorption by SOD, the adsorption characteristics are investigated experimentally at 25, 35 and 45 °C using the more common Langmuir isotherm model and Freundlich isotherm model.

#### 3.6.1. Langmuir Isotherm

The Langmuir isotherm model assumes that adsorption occurs at a specific uniform location within the adsorbent and is applicable to monolayer adsorption containing a finite number of identical adsorption sites. The isotherm equation is shown below (Equation (14)).
(14)$$\frac{C_{e}}{Q_{e}}=\frac{1}{bQ_{m}}+(\frac{1}{Q_{m}})C_{e}$$
wherein: *Q_e_* is the unit adsorption amount at equilibrium (mg/g). *C_e_* is the concentration at adsorption equilibrium (mg/L). *Q_m_* is the maximum monolayer adsorption amount (mg/g). *b* is the adsorption equilibrium constant (L/mg) associated with the free energy of adsorption.

The dimensionless parameter *R_L_* can represent the basic characteristics of the Langmuir isotherm. As shown in Equation (15).
(15)$$R_{L}=\frac{1}{1+bC_{0}}$$
wherein: *C_0_* is the initial concentration (mg/L); *b* is the adsorption equilibrium constant (L/mg).

The value of the dimensionless parameter *R_L_* between 0 and 1 represents favorable adsorption, while *R_L_* > 1 represents unfavorable adsorption, *R_L_* = 1 represents linear adsorption and *R_L_* = 0 represents irreversible adsorption.

The adsorption isotherm experimental data are measured at different concentrations of solutes (ammonium and phosphate), and at different temperatures. The linearized equations of the Langmuir model shown in Figure 12 are obtained by plotting *C_e_*/*Q_e_* against *Q_e_*. As can be seen from the parameters in Table 4, for both ammonium and phosphate, Langmuir isothermal model determination coefficients R^2^ are above 0.99 and 0.98, which can be well fitted to the linear equation. The higher coefficient of determination indicates that the experimental data are in good agreement with the isotherm parameters. This confirms that the adsorption of ammonium and phosphate on SOD is monolayer adsorption [39]. The Langmuir constants *b* is 0.0462 L/mg (25 °C), 0.0858 L/mg (35 °C) and 0.0979 L/mg (45 °C) for ammonium and 0.0180 L/mg (25 °C), 0.0610 L/mg (35 °C) and 0.0683 L/mg (45 °C) for phosphate. The lower *b* values indicate that SOD has a higher affinity for the adsorption of both. The dimensionless parameters *R_L_* for ammonium and phosphate adsorption by SOD ranges from 0.024 to 0.68 and 0.046 to 0.84 (0 < *R_L_* < 1), which indicates that the adsorption of synthetic materials is easier to occur for the removal of ammonium and phosphate.

#### 3.6.2. Freundlich Isotherm

As an empirical equation, the Freundlich isotherm describes the inhomogeneous surface energy equation and is applicable to non-homogeneous systems. The form is as in Equation (16).
(16)$$\log Q_{e}=\log K_{f}+n_{f}\log C_{e}$$
where in: *Q_e_* and *C_e_* are the same as in the Langmuir isotherm model; *Q_m_* is the maximum monolayer adsorption amount (mg/g); *K_f_* is the adsorption coefficient (L/mg) indicating the relative adsorption capacity. *n_f_* is the adsorption index indicating the adsorption strength.

Freundlich adsorption isotherms for ammonium and phosphate are obtained by plotting the linear relationship between log*C_e_* and log*Q_e_* (Figure 13). From Table 4, we can see that the coefficient of determination of the Freundlich isothermal model for both, R^2^ > 0.99. *K_f_* values range between 0.2491 to 0.2785 and 0.2491 to 0.2725, *n_f_* ranges between 0.1 and 1, which further illustrates that SOD was beneficial for ammonium and phosphate adsorption.

Combining the adsorption isotherms obtained from the two adsorption models, it was found that the adsorption process for ammonium and phosphate is not a single monolayer adsorption [16]. For adsorbed ammonium, the coefficients of determination R^2^ are all greater than 0.99, a higher fit to both isothermal models. The maximum adsorption capacities obtained from the Langmuir adsorption isotherm, 17.29 mg/g (25 °C), 28.69 mg/g (35 °C) and 30.21 mg/g (45 °C), are identical to the actual measured maximum adsorption capacities of 26.26 mg/g (25 °C), 28.57 mg/g (35 °C) and 30.53 mg/g (45 °C). For phosphate, the Freundlich model outperforms the Langmuir model with larger R^2^ values and better correlation. The Freundlich model for phosphorus adsorption equilibrium constant *K_f_* increases with increasing temperature, indicating that the affinity of SOD for phosphate adsorption increases with increasing temperature.

### 3.7. Adsorption Thermodynamics

The thermal effect in the adsorption process is called the heat of adsorption, and the magnitude of the heat of adsorption reflects the change in the strength of the adsorption. The adsorption process is determined by measuring multiple adsorption isotherms in each temperature range and obtaining the equivalent heat of adsorption by calculation. The thermodynamic parameters of ammonium and phosphate adsorption by SOD are calculated based on the constant b obtained from the Langmuir isotherm model, wherein includes the enthalpy change of adsorption (Δ*H*), the entropy change (Δ*S*) and the change of Gibbs free energy (Δ*G*). It is calculated by Equations (17) and (18).
(17)$$\ln b=\frac{\Delta S}{R}-\frac{\Delta H}{RT}$$
(18)ΔG=ΔH−TΔS
where in: ∆*G*, ∆*H* denote Gibbs free energy change and enthalpy change (KJ/mol); ∆*S* denotes the entropy change during the reaction [J/(mol·K)]; *b* is the Langmuir model constant (mg/L) at different temperatures.

The details of the calculated thermodynamic parameters are shown in Table 5. At 25, 35 and 45 °C, the free energy obtained by adsorption of ammonium is −19.22 KJ·mol^−1^, −21.09 KJ·mol^−1^ and −22.95 KJ·mol^−1^, and the free energy changes obtained by adsorption of phosphate are −0.03 KJ·mol^−1^, −0.12 KJ·mol^−1^ and −0.22 KJ·mol^−1^. The free energy changes (∆*G*) of both ammonium and phosphate are negative and decrease with increasing temperature in the experimentally defined temperature range. This indicates that the adsorption is a spontaneous process and confirms that temperature increase is favorable for the adsorption reaction to proceed, which is consistent with the conclusion of the adsorption isotherm mentioned above. For ammonium and phosphate, the enthalpy change (∆*H*) values are 36.37 and 2.77 KJ·mol^−1^, with positive values, indicating that the adsorption of both is a heat-absorbing reaction [10,39]. The entropy change values (∆*S*) remain positive values for ammonium and phosphate (186.56, 9.39 J·mol^−1^·K^−1^), which indicates that the increase in confusion at the solid-liquid reaction interface in adsorption improves the removal for ammonium and phosphate.

### 3.8. Desorption and Regeneration Experiment

First, in order to consider whether the adsorbent has a desorption phenomenon under normal conditions, the saturated SOD is poured into deionized water to determine the solute content in the solution. As can be seen from Table 6, the desorption rate of nitrogen and phosphorus are relatively low. Therefore, the use of SOD as adsorbent to adsorb nitrogen and phosphorus has a good solidification effect and will not leach out under natural conditions to cause secondary pollution.

When the leaching of the adsorbed saturated SOD was performed using desorbent (Figure 14), it was found that the desorption rate of nitrogen was 85.6%, which was better than that of phosphorus, mainly due to the reversible exchange process between Na^+^ and NH_4_^+^. During the adsorption reaction, the regeneration rate of adsorbed ammonia nitrogen was 83.6%, and the regeneration rate of adsorbed phosphorus was 42.3%. The reduction in regeneration effect may be due to the destruction of the internal structure of the material by the desorption liquid.

## 4. Conclusions

In this paper, sodium aluminosilicate of sodium squared stone type has been successfully prepared by one-step hydrothermal alkali dissolution desilication using fly ash as raw material. The adsorption performance of SOD in waste water solution has been studied, and its adsorption mechanism has been investigated.

The silica fraction in the raw ash was adequately debonded in a hot alkaline solution at 80 °C. Heated to 90 °C and stirred rapidly, the glassy phase in fly ash is successfully converted to hydrated sodium aluminosilicate. The new formed crystalline phase overlies the undissolved by-product mullite and shows a cluster-type petal shape. Therefore, the surface of the SOD is uneven and has a large number of holes. Combined with the characterization analysis, the cation exchange capacity is 136.5 cmol/kg and the specific surface area is 60.54 m^2^/g, indicating that SOD has the potential to be used as an adsorbent.

Through batch adsorption experiments, the influence of the amount of adsorbent, contact time and initial pH value on the adsorption of ammonium ions and phosphate by SOD has been explored. When C_P0_ = 10 mg/L and C_N0_ = 50 mg/L, the optimal adsorption conditions for SOD to adsorbed ammonium and phosphorus are 4 g/L, *t* = 90 min, and pH = 8.

In the exploration for the initial concentration of solute. It is found that the release of Ca^2+^ from the SOD is induced at higher ammonium concentrations. Ca^2+^ produced more calcium phosphate precipitates with phosphate in solution, thus increasing the phosphorus removal.

The synthetic material SOD has a good curing effect on ammonium and phosphorus. The adsorbed SOD is put into deionized water and stirred, repeated three times, there is no obvious desorption phenomenon and secondary pollution is avoided. It has good regeneration performance after desorption of SOD saturated with adsorbed ammonium, and the regeneration rate reaches 83.6%.

In order to further investigate the adsorption mechanism. The quasi-secondary kinetic model is found to be the best way to describe the simultaneous adsorption of ammonium and phosphorus on SOD, suggesting that the adsorption process is accomplished synergistically by physical and chemical adsorption. The Freundlich isotherm model is superior to the Langmuir isotherm data, so adsorption in the non-homogeneous layer needs to be considered. Thermodynamic evaluation shows that the adsorption processes of both ammonium and phosphorus on SOD are heat-absorbing and spontaneous.

With the advantages of low cost and high adsorption capacity, SOD has shown great advantages and potential as an adsorbent to remove nutrients from wastewater.

## Figures and Tables

**Figure 1 materials-14-02741-f001:**
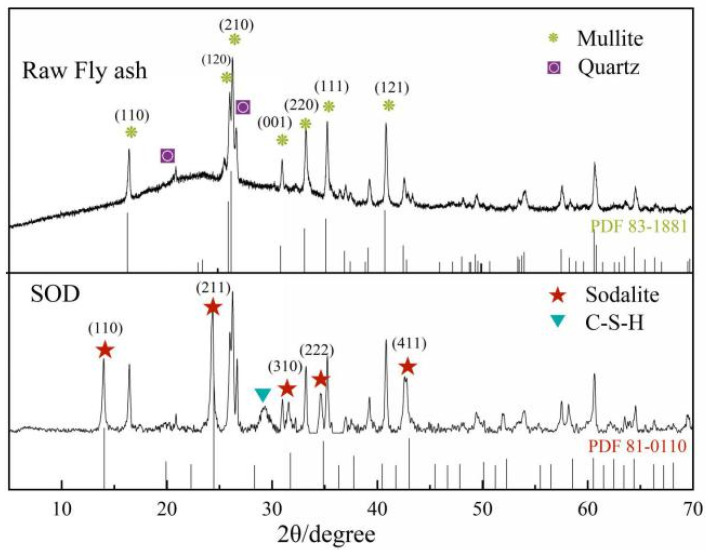
XRD patterns of raw material (FA) and synthetic product (SOD).

**Figure 2 materials-14-02741-f002:**
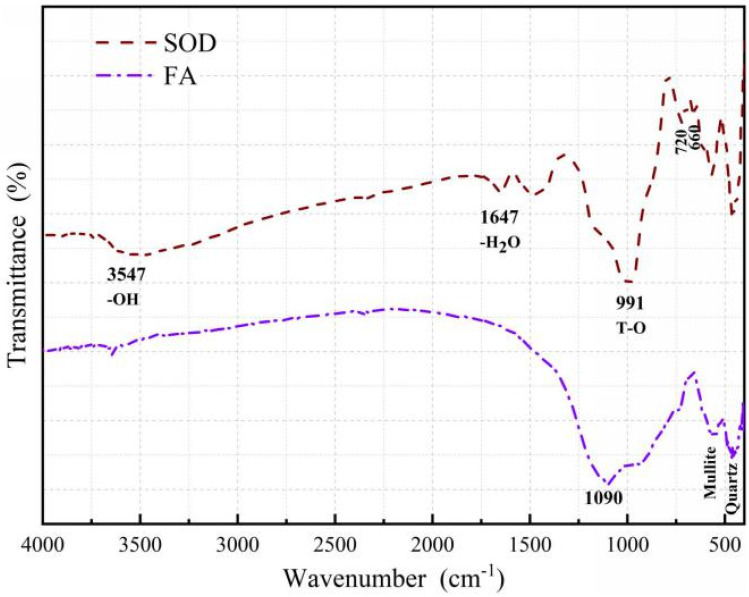
FTIR spectra of raw material (FA) and alkali-soluble desiliconization products (SOD). T-O stands for T-Si or T-Al.

**Figure 3 materials-14-02741-f003:**
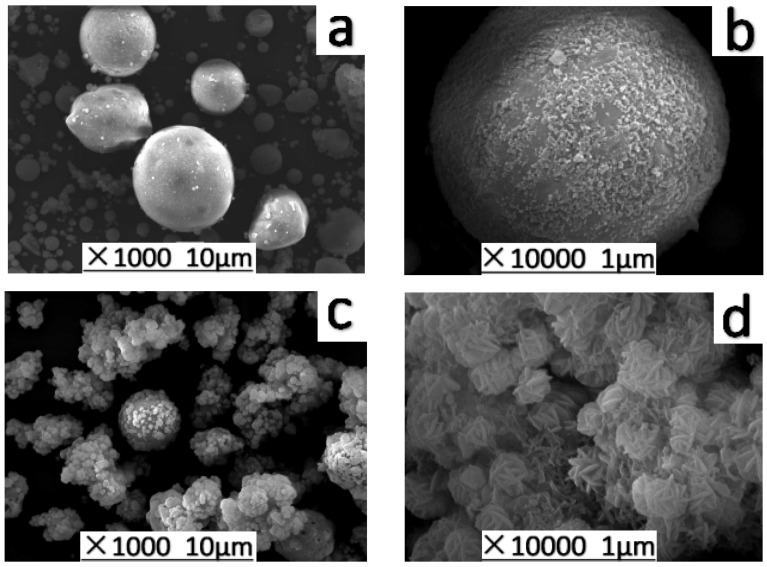
SEM images: (**a**,**b**) FA; (**c**,**d**) SOD.

**Figure 4 materials-14-02741-f004:**
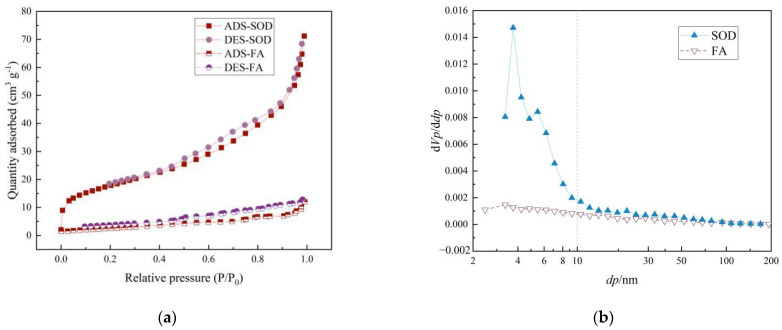
BET test results of raw material FA and synthetic product SOD (**a**) N_2_ physisorption isotherm and (**b**) pore size distribution calculated by BJH method.

**Figure 5 materials-14-02741-f005:**
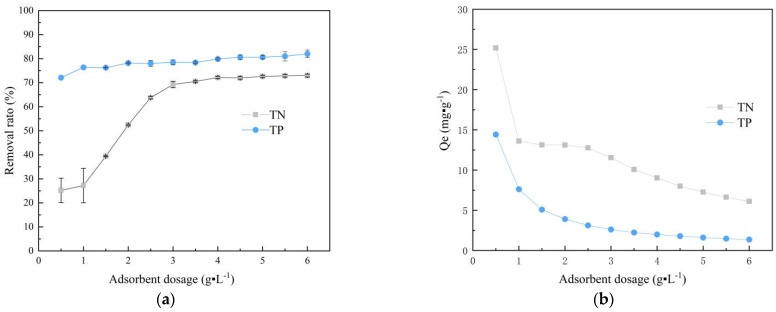
Effect of SOD dosage on simultaneous ammonium and phosphorus removal. (**a**) Change graph of removal rate with increase in dosage, (**b**) change graph of unit adsorption capacity with increasing dosage (C_P0_ = 10 mg/L; C_N0_ = 50 mg/L; pH = 7.0; *T* = 25 °C; *t* = 90 min).

**Figure 6 materials-14-02741-f006:**
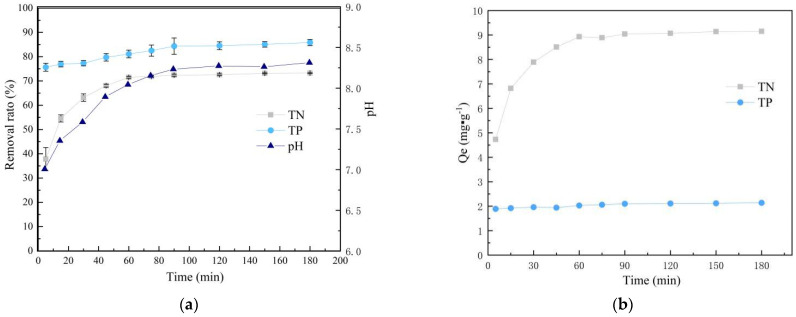
Effect of adsorption time on ammonium and phosphorus removal. (**a**) Change graph of removal rate with increasing contact time, (**b**) change graph of unit adsorption capacity with increasing contact time (C_P0_ = 10 mg/L; C_N0_ = 50 mg/L; pH = 7.0; *T* = 25 °C; adsorbent dosage = 4 g/L).

**Figure 7 materials-14-02741-f007:**
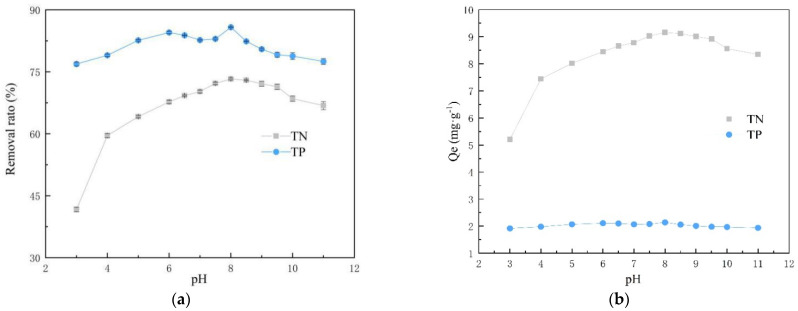
Effect of solution pH value on simultaneous ammonium and phosphorus removal. (**a**) Change graph of removal rate with increasing pH value, (**b**) change graph of unit adsorption capacity with increasing pH value (C_P0_ = 10 mg/L; C_N0_ = 50 mg/L; pH = 7.0; *T* = 25 °C; adsorbent dosage = 4 g/L; *t* = 90 min).

**Figure 8 materials-14-02741-f008:**
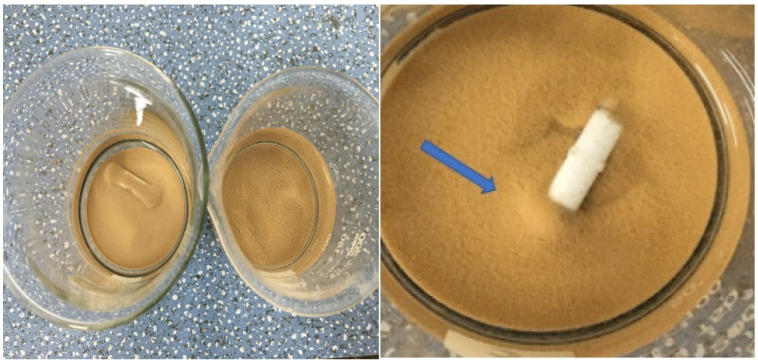
The image of the adsorbent forming flocculation after adsorption and precipitation.

**Figure 9 materials-14-02741-f009:**
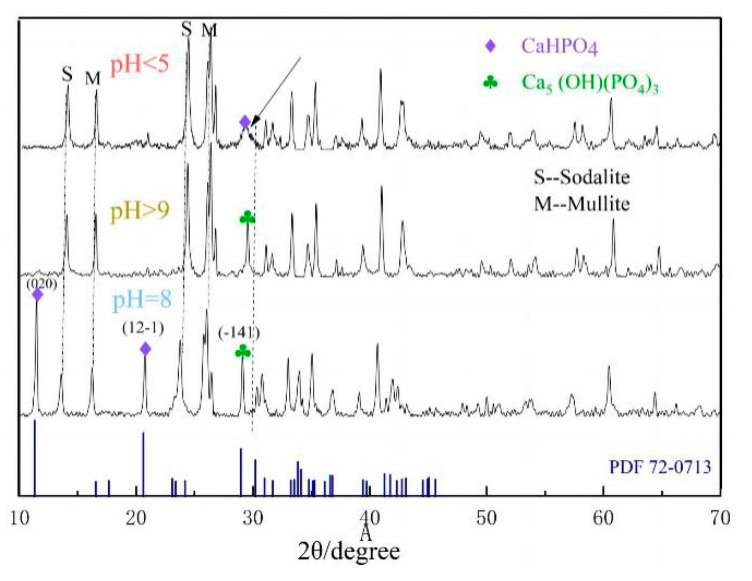
XRD patterns of adsorbed sediments at different pH values.

**Figure 10 materials-14-02741-f010:**
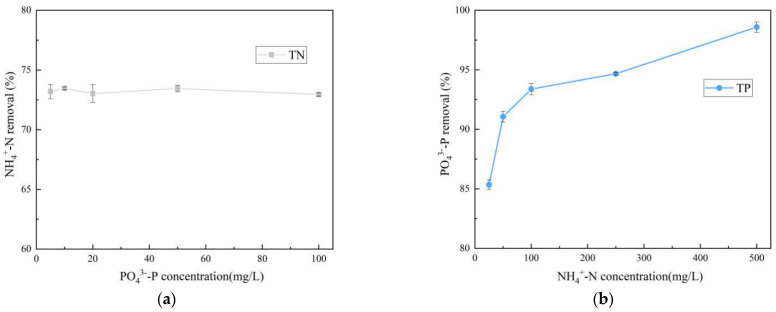
Effect of initial concentration of solution on simultaneous ammonium and phosphorus removal. (**a**) C_N_ = 50 mg/L, C_P_ = 5–100 mg/L; (**b**) C_P_ = 10 mg/L, C_N_ = 25–500 mg/L (pH = 8.0; *T* = 25 °C; adsorbent dosage = 4 g/L; *t* = 90 min). Three sets of parallel tests are taken, and the error results are expressed in the form of error bars.

**Figure 11 materials-14-02741-f011:**
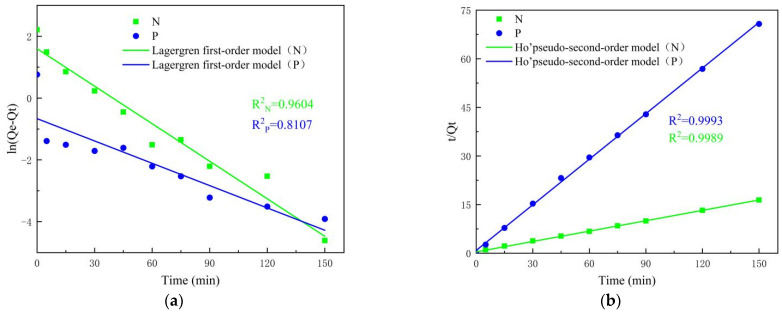
Linearization of ammonium and phosphate adsorption kinetic. (**a**) Lagergren’s first-order model; (**b**) Ho’s pseudo-second-order model.

**Figure 12 materials-14-02741-f012:**
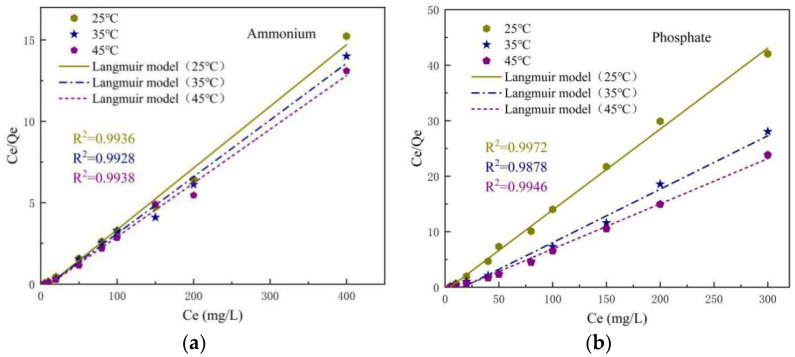
Langmuir isotherm plots for the adsorption of ammonium (**a**) and phosphate (**b**) onto the SOD.

**Figure 13 materials-14-02741-f013:**
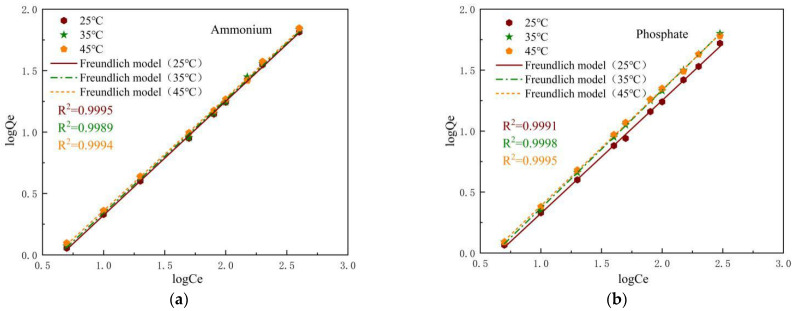
Freundlich isotherm plots for the adsorption of ammonium (**a**) and phosphate (**b**) onto the SOD.

**Figure 14 materials-14-02741-f014:**
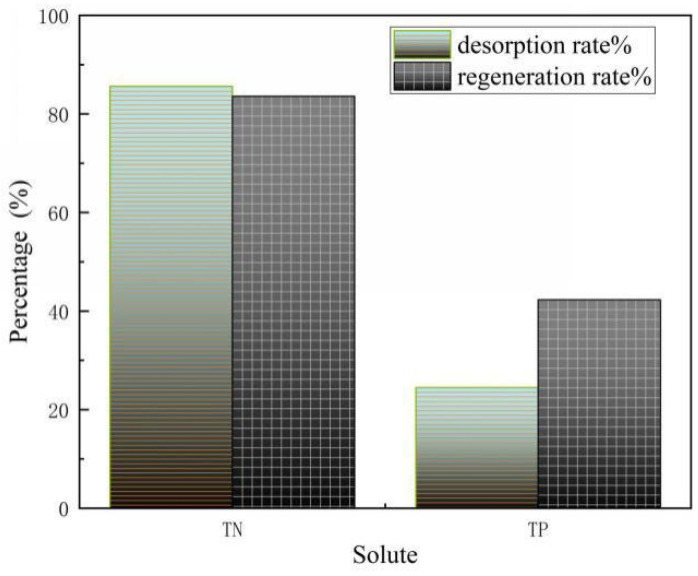
Desorption and regeneration of nitrogen and phosphorus by saturated SOD adsorption.

**Table 1 materials-14-02741-t001:** Chemical constituent of FA and SOD (wt%).

Sample	Chemical Composition (wt%)						
	SiO_2_	Al_2_O_3_	CaO	Fe_2_O_3_	MgO	Na_2_O	LOI
FA	41.31	34.30	9.41	7.00	1.54	0.31	1.33
SOD	31.11	30.87	7.15	5.74	1.06	11.21	--

**Table 2 materials-14-02741-t002:** Chemical constituents of SOD and adsorbed sediment slag (wt%).

Sample	Chemical Composition (wt%)							
	SiO_2_	Al_2_O_3_	CaO	Na_2_O	K_2_O	MgO	Fe_2_O_3_	LOI
1 (C_N_ = 25 mg/L)	32.09	31.53	7.85	10.59	0.28	1.06	5.74	--
2 (C_N_ = 50 mg/L)	32.52	31.55	7.96	9.38	0.29	1.14	5.82	--
3 (C_N_ = 100 mg/L)	32.73	32.04	7.97	9.34	0.30	1.14	5.86	--
4 (C_N_ = 250 mg/L)	32.55	31.96	7.96	9.30	0.29	1.15	5.89	--
5 (C_N_ = 500 mg/L)	32.61	31.83	8.05	9.22	0.30	1.16	5.88	--

**Table 3 materials-14-02741-t003:** Kinetic parameters.

Adsorbate	*Q_e_* (mg/g)	Lagergren First-Order			Ho’s Pseudo-Second-Order		
		R^2^	*K*_1_ (min^−1^)	*Q_e_*_,1_ (mg/g)	R^2^	*K*_2_ [g/(mg min)]	*Q_e_*_,2_ (mg/g)
Ammonium	9.15	0.9604	0.0405	4.9499	0.9989	2.5056	9.3832
Phosphate	2.14	0.8107	0.0241	0.5147	0.9993	1.5163	2.1463

*Q_e_* is the experimentally measured adsorption amount at adsorption equilibrium; *Q_e_*_,1_, *Q_e_*_,2_ are the calculated values of kinetic model fitting.

**Table 4 materials-14-02741-t004:** Langmuir and Freundlich isotherm parameters.

Adsorbate	Temp (°C)	Langmuir Parameters			Freundlich Parameters		
		*Q_m_* (mg/g)	*b* (L/mg)	R^2^	*K_f_* (mg·g^−1^)	*n_f_*	R^2^
Ammonium	25	17.2891	0.0462	0.9935	0.2491	0.9274	0.9995
	35	28.6944	0.0858	0.9928	0.2622	0.9228	0.9989
	45	30.3122	0.0979	0.9938	0.2785	0.9163	0.9994
Phosphate	25	6.8573	0.0180	0.9972	0.2491	0.9260	0.9991
	35	10.3950	0.0610	0.9878	0.2516	0.9676	0.9998
	45	12.2911	0.0683	0.9946	0.2725	0.9527	0.9995

**Table 5 materials-14-02741-t005:** Thermodynamic parameters.

Adsorbent	∆*G* (KJ·mol^−1^)			∆*H* (KJ·mol^−1^)	∆*S* (J·mol^−1^·K^−1^)
	298 K	308 K	318 K		
Ammonium	−33.64	−38.77	−43.91	119.48	531.81
Phosphate	−13.89	−16.13	−18.38	52.98	224.39

**Table 6 materials-14-02741-t006:** Desorption in water of SOD saturated with adsorbed nitrogen and phosphorus.

Times	The Desorption Rate (%)	
	Ammonium	Phosphate
1	0.2	0.1
2	0.2	0
3	0.1	0

## Data Availability

Data is contained within the article.

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
