# Peer review of "Hydrothermal Synthesis of Sodalite-Type N-A-S-H from Fly Ash to Remove Ammonium and Phosphorus from Water"

_materials, 2021, doi:10.3390/ma14112741_

Round 1

Reviewer 1 Report

Dear Editor,

The pH of the environment for the reactions of 5-8 must be unambiguously determined. The Lagergren's first order model must be unambiguously presented. English must be carefully corrected!

With kind regards

Author Response

Response to Reviewer 1 Comments

Point 1: The pH of the environment for the reactions of 5-8 must be unambiguously determined. The Lagergren's first order model must be unambiguously presented. English must be carefully corrected!

Response 1: The authors would like to thank Reviewer 1 for comprehensive criticism and suggestions. We consider the environmental factors of the reaction pH value has been strictly controlled between 3-11. The Lagergren's first-order model has been clearly proposed in the article. English language has been carefully revised.

Reviewer 2 Report

The authors have carried out an interesting study of the synthesis of a material based on hydrated sodium aluminosilicate and have used it in the adsorption of ammonium and phosphorus. The manuscript can be published in Materials, but some aspects should be revised.
1. In the introduction, the authors should show a table with the most relevant results obtained by other authors in the adsorption of ammonium and phosphorus with other materials.
2. In the characterization section, the authors should show the image of the nitrogen adsorption-desorption isotherm of the synthesized materials, and indicate the pore volume and the pore size distribution graph.
3. A material regeneration study should be developed to check the adsorption-desorption cycles valid for the process.
4. Determination of the zeta potential of the material is recommended to further study the pH developed.
5. A review of the English language is necessary throughout the manuscript.

With the suggested changes, the manuscript could be accepted in Materials.

Author Response

Response to reviewer 2 comments

The authors have carried out an interesting study of the synthesis of a material based on hydrated sodium aluminosilicate and have used it in the adsorption of ammonium and phosphorus. The manuscript can be published in Materials, but some aspects should be revised.

The authors would like to thank Reviewer 2 for the critical and constructive comments. We have made careful revisions based on the suggestions, and hope to get your approval.
Point 1: In the introduction, the authors should show a table with the most relevant results obtained by other authors in the adsorption of ammonium and phosphorus with other materials.
Response 1: Thank you for your critical and detailed advice. We have expanded the content on the introduction part accordingly.

Number

Authors

Name of synthetic material

Adsorption of ammonium (max)

Adsorption of phosphate (max)

Reference

1

Ji et al

SF-Z,

CF-Z

12.97mg/L, 87.41mg/L

[1]

2

You et al

Ze-Ca,

Ze-Mg1

Ze-Mg2

123.1mg/L,

55.2mg/L

32.3mg/L

119.5mg/L,

60.5mg/L

23.9mg/L

[2]

3

Li et al

LMZ

64.1mg/L

[3]

  1. Ji, X. D.;Zhang, M. L.; Ke, Y. Y.; Song, Y. C. Simultaneous immobilization of ammonium and phosphate from aqueous solution using zeolites synthesized from fly ashes. Water. Sci. Technol. 2013, 67, 1324–1331, doi:10.2166/wst.2013.690. 
  2. You, X.;Valderrama, C.; Cortina, J. L. Simultaneous recovery of ammonium and phosphate from simulated treated wastewater effluents by activated calcium and magnesium zeolites. J. Chem. Technol. Biot. 2017, 92, 2400–2409, doi:10.1002/jctb.5249.
  3. Li, X.;Xie, Q.; Chen, S.; Xing, M.; Guan, T.; Wu, D. Inactivation of phosphorus in the sediment of the Lake Taihu by lanthanum modified zeolite using laboratory studies. Environ. Pollut. 2019247, 9–17, doi:1016/j.envpol.2019.01.008. 

Piont 2: In the characterization section, the authors should show the image of the nitrogen adsorption-desorption isotherm of the synthesized materials, and indicate the pore volume and the pore size distribution graph.

Response 2: Thank you for your advice. We reorganized the content and added specific results of the BET test in terms of characterizing materials in the text. In the article, we added N2 adsorption/desorption isotherms and pore size distribution maps, and compared the raw materials and the synthesized products.

Piont 3: A material regeneration study should be developed to check the adsorption-desorption cycles valid for the process.
Response 3: Your suggestion is very meaningful. We have carried out certain supplementary experiments on desorption/regeneration of saturated adsorption materials. In the article, we introduced in detail the selection and understanding of the desorption process, and carried out regeneration experiments.

Piont 4: Determination of the zeta potential of the material is recommended to further study the pH developed.

Response 4: According to the suggestion, we will understand the experiment related to zeta potential. Because the experimental conditions are not met, the zeta potential cannot be measured temporarily.

Piont 5: A review of the English language is necessary throughout the manuscript.

Response 5: Thank you very much for your suggestion. In response to the suggestions, we have made certain changes to the English grammar.

Reviewer 3 Report

I would like to congratulate the authors for the very good manuscript written based on high-quality analytical results. I would suggest they change present tense to past tense in the results and discussion section. 

The abstract is very good, but it should be carefully checked for grammar (line 16: for the removal of; 19: show; 21: showed). 

Line 45: what is FA? Afterwards, fly ash should not be used, but FA instead. 

Line 125: analytical quality check should be presented. 

Line 169: showed

171: mineral names in small letters, not capital ones. 

Table 1: give data in 3 significant figures, not more (e.g. 41.3, 31.1, 0.31, etc.). 

Figure 3: please, increase the font.

236: 69.3%, 78.5%, 82.0%.

273: typing error, please make an indent (5 min, not 5min).

Figure 8 is blurred, try to make it more clear (sharp).

Figure 5: what are TN and TP? Please, provide explanations (descriptions) of abbreviations where needed. 

Author Response

Response to Reviewer 3 Comments

Point 1: The abstract is very good, but it should be carefully checked for grammar (line 16: for the removal of; 19: show; 21: showed).

Response 1: The authors would like to thank Reviewer 3 for the critical and constructive comments. We have checked some grammatical errors and made corrections in the text.

Piont 2: Line 45: what is FA? Afterwards, fly ash should not be used, but FA instead.

Response 2: Thank you for your advice. We have made corrections in the text and defined the acronym "FA" for fly ash.

Piont 3: Line 125: analytical quality check should be presented.

Response 3: Thank you for your advice. As part of the characterization test was sent for inspection, no analytical quality check was given. We will pay attention to the quality analysis report.

Piont 4: Line 169: showed

Response 4: Thank you for your advice. We have modified the original text.

Piont 5: 171: mineral names in small letters, not capital ones.

Response 5: Thank you very much for your suggestion. We have modified the way of writing "mullite" and "quartz".

Piont 6: Table 1: give data in 3 significant figures, not more (e.g. 41.3, 31.1, 0.31, etc.).

Response 6: Thank you for your advice. We have bolded the salient data in Table 1 of the main text.

Piont 7: Figure 3: please, increase the font.

Response 7: Thank you for your advice. We have modified Figure 3 in the original text

Piont 8: 236: 69.3%, 78.5%, 82.0%.

Response 8: Thank you very much for your suggestion. We have modified in the text.

Piont 9: 273: typing error, please make an indent (5 min, not 5min).

Response 9: Thank you for your careful review, we have revised the article.

Piont 10:Figure 8 is blurred, try to make it more clear (sharp).

Response 10: Thank you for your advice. We have replaced Figure 8 with a clear image

Piont 11: Figure 5: what are TN and TP? Please, provide explanations (descriptions) of abbreviations where needed.

Response 11: TN and TP represent the total ammonium and total phosphorus in the solution, respectively. We have defined abbreviations in the main text.

Reviewer 4 Report

The abstract can be supported by reporting some data. 

Page 7. line 227: "The linear relationship between the unit adsorption 227 capacity (Fig 4b) and the removal rate (Fig 4a)". It is not linear. Please clarify.

Fig 4 & 5. Why does the sharp increase for TN removal is observed but not in the case of TP?

Fig 7. It is not clear in the figure what to be shown. 

Author Response

Response to Reviewer 4 Comments

Point 1: The abstract can be supported by reporting some data. 

Response 1: The authors would like to thank Reviewer 4 for the critical and constructive comments. We have added some data analysis to the summary to make the content richer.

Piont 2: Page 7. line 227: "The linear relationship between the unit adsorption 227 capacity (Fig 4b) and the removal rate (Fig 4a)". It is not linear. Please clarify.

Response 2: Thank you for your advice. There is no linear relationship between the unit adsorption capacity and the removal rate. This is our expression error and has been revised in the text.

Piont 3: Fig 4 & 5. Why does the sharp increase for TN removal is observed but not in the case of TP?

Response 3: In Figs. 4 and 5, the significant removal of TP but not TP is mainly due to the different adsorption rates of both. With the increase of the adsorbent dosage and the extension of the reaction time, the adsorption law of TN is slow adsorption and slow equilibrium, while the adsorption law of TP is fast adsorption and slow equilibrium, so the curve of TP looks relatively flat.

Piont 4: Fig 7. It is not clear in the figure what to be shown. 

Response 4: Thank you for your advice. For Figure 7, we mainly want to express the precipitation and flocs formed when adsorbing high concentrations of phosphate. As can be seen in the figure, the gray matter is glued together.

In the above figure, the beaker on the left is the SOD in deionized water, and the beaker on the right is the SOD after phosphate adsorption and settling. It can be seen that obvious flocs are formed after adsorption, and there are sediments in the flocculation.

Reviewer 5 Report

The manuscripts deals with products made from the pollutant, fly ash, to combat eutrophication of wastewater. Eutrophication is a very critical issue seen around the world. This article shows possible insight to for a new way that this buildup of ammonium and phosphorus can be effectively removed. This article shows that the use of sodalite-type, in the sodium aluminosilicate crystalline phase, has to capacity for adsorption of the harmful nutrients that cause eutrophication.

Fly ash is used in many other papers for wastewater treatment, primarily through zeolite formation reducing the originality / novelty of the work, but this seems to be the first to use NASH specifically for this purpose.

The manuscript is difficult to follow and many grammatical errors and issues with tense that must be extensively corrected.  The authors must include more detailed figure captions. They do not describe what the figure actually is or the relevancy. There are problems with the structure of the sentences and missing explanations for the reader. The abbreviations are used without declaration or previous declaration.  A key requirement of Materials is that the materials section be detailed for other scientists to reproduce. Some analytical methods, such as ICP, are not included in the materials section. The term sputter coating (page 3 line 120) should be used with details on the instrument, time, and layer depth.  Results like BET are not properly presented and discussed. The authors, although they do not specify mass or volumes used, provide ratios for preparation. Also, the authors do not mention why they chose 25, 35, and 45°C as temperatures to test. No bragg  peak assignments are provided. Citations (41) are primarily all within the last 20 years except one.

Page 1, Line 25 and 42,  use of the word nutrients is not appropriate, should be contaminants.

Page 1, Line 31, what water column?

Page 1, line 31-33: The tense of the sentence is mixed, needs to be reworded. What is too high a level of these ions?

Page 1, line 35 keep consistent order of ammonium before phosphate throughout the manuscript and do not switch back and forth.

Page 1, Line 35, change “is” to “are”

Page 2, Line 45, need to define FA as fly ash

Page 2, Line 46-49, sentence should be rewritten: “Because of the complex structure of FA and the toxic elements resulting from random superimposed influence of various factors such as the nature of raw materials and formation conditions, it limits its sustainable resource utilization pathway.” ; unsure about the use of random and superimposed in Line 47

Page 2, Line 49-50, “It was found that the chemical composition of fly ash is extremely similar to zeolite, allowing for a high value utilization of fly ash.”

Page 2, line 73 define secondary pollution.

Page 2, Line 75, remove “for resource utilization” because it was in the previous line

Page 2, Line 78, “experiences”

Page 2, Line 81, “post-modification”

Page 2, Line 83 change "shows an exploration" to examines

Page 2, Line 85, “analyses of the adsorbed precipitates”

Page 2, Line 91 define AR

Page 3, Line 99, what constant weight? What’s the starting mass of FA?

Page 3, Line 100, change “function” to “purpose”

Page 3, line 105: This sentence needs to be reworded “stirred at an even speed of 80°C”. I am interested to know at what speed this was performed.

Page 3,  line 113: should be XRPD X-ray powder diffraction.

Page 3, line 130: What was the constant temperature used during stirring for the batch experiment? What are AR and GR?

Page 3, line 132. Please define Cp and Cn. Also, define any other abbreviations in the text.

Page 3, Line 133, rated concentration? Written? what do the authors mean by preconfigured? Was a pH meter used?

Page 3, line 142. Explain why the initial ammonium and phosphate concentrations are configured to be 50 mg N/L and 10 mg P/L. Only the reference is not enough. Explain that  N represents ammonium and P phosphate.

Page 3, Line 145 – Page 4, Line 148, sentences are not clear

Page 4, line 160-164 is Qe unitless? How does Ce relate to  Ct? How do the units cancel when C is in mg/L and W in g? 

Page 4, line 169 results show

Page 4, line 171 & 174 should be broad not bread peak

Page 4 equations need states of matter such as 

Page 5, line 177: add more description to the caption. Add the space between intensity and AU- correct this in all y-axes of the figures.

Page 5, Line 178, Figure 1 needs a few sentences of description (caption).

Page 5, line 179 Please articulate and better explain the sentence: The chemical composition of the synthesized product is basically similar to that of 179 the original ash (Table 1), and the molecular ratio of the formed sodium aluminosilicate 180 solution is on the high side.

Page 6, line 205. ‘’The morphological analysis of fly ash’’ is not properly correct since the authors are only showing images and not performing any types of analysis. Comment on the surface area changes from SEM images.

Page 6 line 210: Define and explain BET. Authors claim they examined the internal pore structure and specific surface area of the materials by N2 adsorption BET by use of a Micro-meritics ASAP 2020 volumetric instrument in the methods. Add more details.

 Page 6, Line 204, Figure 2 needs a few sentences of description (caption). Also needs to be bigger because it is difficult to read. The legend says MFA and FA, but the caption states FA and SOD. 

Page 6, Line 209, Figure 3 needs a few sentences of description (caption). Also should be A, B, C, D. Don't use blue on a black background.

Page 6, Line 212, define CEC

Page 7, lines 227. Why is b discussed before a? Correct this. How authors calculated the linear relationship between the unit adsorption capacity and the removal rate with dose should be added here.

Page 7, Line 228-229, this is not a complete sentence. There is no need to state again “from the figure” after the sentence before it ended with “can be seen in the figure”

Page 7 line 232-234: This sentence is confusing. You state that when the dosage is lower than 3 g·L the removal rates increase as the dosage increases. Do you mean that 3 g·L is the max does to see an increase uptake of ammonium and phosphate or is that when first start to see an uptake?

Page 7, Line 235, missing L-1 in 6g*L

Page 7, Line 237, “decreases”

Page 7, line 250. Please define TP TN. Also I wonder if there is a better way to graphically represent the 2 graphs since they are plotting on the same x-axis.

Figure 4: the legend improvements. Please specify what is TN and TP. What is the error?

Page 8 line 269: please specify what is “the material” is this SOD?

Figure 5: once again please clarify the legend.

Page 10, Line 310, Figure 7 needs a few sentences of description (caption). What is the blue arrow pointing to and why is this important? A scale bar would have been nice.

Page 10, Line 310, Figure 8 needs a few sentences of description (caption). Difficult to read.

Page 10, Line 332, ICP is not mentioned in methods section. OES or MS as detector? Also needs to be defined.

Page 11, line 349-350 subscript solution is misspelled. Assign states of matter to the equation.

Page 11 line 333: Which ICP is used OES or MS? Clarify.

Page 14, Line 422, Figure 11 needs a few sentences of description (caption).

Page 15, Line 310, Figure 7 needs a few sentences of description (caption).

Page 16, Line 472, why suddenly change from Celsius to Kelvin?

Page 16, Line 487, cmol?

Page 16, Line 489, not a complete sentence

Page 16, Line 494, not a complete sentence

Page 16, Line 498, not a complete sentence

Page 16 section 4: Use of bullet points for the  conclusion is not appropriate. Change this to a paragraph format.

Another general comment is to add a section indicating how you calculate the fitting models (software used, methods etc)

Page 18, Line 605-606, citation format not consistent

Page 18, Line 607-608, citation format not consistent

Author Response

Response to Reviewer 5 Comments

The manuscripts deals with products made from the pollutant, fly ash, to combat eutrophication of wastewater. Eutrophication is a very critical issue seen around the world. This article shows possible insight to for a new way that this buildup of ammonium and phosphorus can be effectively removed. This article shows that the use of sodalite-type, in the sodium aluminosilicate crystalline phase, has to capacity for adsorption of the harmful nutrients that cause eutrophication.

Fly ash is used in many other papers for wastewater treatment, primarily through zeolite formation reducing the originality / novelty of the work, but this seems to be the first to use NASH specifically for this purpose.

The manuscript is difficult to follow and many grammatical errors and issues with tense that must be extensively corrected. The authors must include more detailed figure captions. They do n ot describe what the figure actually is or the relevancy. There are problems with the structure of the sentences and missing explanations for the reader. The abbreviations are used without declaration or previous declaration. A key requirement of Materials is that the materials section be detailed for other scientists to reproduce. Some analytical methods, such as ICP, are not included in the materials section. The term sputter coating (page 3 line 120) should be used with details on the instrument, time, and layer depth. Results like BET are not properly presented and discussed. The authors, although they do not specify mass or volumes used, provide ratios for preparation. Also, the authors do not mention why they chose 25, 35, and 45°C as temperatures to test. No bragg peak assignments are provided. Citations (41) are primarily all within the last 20 years except one.

The authors would like to thank Reviewer 3 for the critical and constructive comments. We have carefully revised the article according to your suggestions one by one. For ICP-OES, BET and other tests, we have made supplements in the article. Regarding the selection of the experimental temperature of 25/35/45℃, on the one hand, it is based on the experience of the predecessors, on the other hand, it is to compound the actual adsorption conditions. The temperature is too low or too high to easily generate energy consumption. We also revise other errors in the article one by one.

Point 1: Page 1, Line 25 and 42, use of the word nutrients is not appropriate, should be contaminants.

Response 1: Thank you for your advice. We have modified in the text

Piont 2: Page 1, Line 31, what water column?

Response 2: Thank you for your advice. We originally wanted to express the meaning of water body, which has been revised in the text.

Piont 3: Page 1, line 31-33: The tense of the sentence is mixed, needs to be reworded. What is too high a level of these ions?

Response 3: Thank you for your advice. We have modified the tense in the text. Originally wanted to describe the eutrophication caused by high ammonium and phosphorus content, but it has been revised in the original text.

Piont 4: Page 1, line 35 keep consistent order of ammonium before phosphate throughout the manuscript and do not switch back and forth.

Response 4: Thank you for your advice. We will pay more attention to the state of matter and avoid such mixed-use errors.

Piont 5: Page 1, Line 35, change “is” to “are”

Response 5: Thank you very much for your suggestion. We have corrected in the text.

Piont 6: Page 2, Line 45, need to define FA as fly ash

Response 6: Thank you for your advice. We have paid attention to the application of abbreviations and define fly ash as FA.

Piont 7: Page 2, Line 46-49, sentence should be rewritten: “Because of the complex structure of FA and the toxic elements resulting from random superimposed influence of various factors such as the nature of raw materials and formation conditions, it limits its sustainable resource utilization pathway.” ; unsure about the use of random and superimposed in Line 47

Response 7: Thank you for your advice. We have rewritten this sentence.

Due to the superimposed influence of various factors such as the nature of raw materials and formation conditions. The resulting FA has a complex composition structure and contains trace toxic elements, which limits the sustainable resource utilization of FA.

Piont 8: Page 2, Line 49-50, “It was found that the chemical composition of fly ash is extremely similar to zeolite, allowing for a high value utilization of fly ash.”

Response 8: Thank you very much for your suggestion. We have rewritten this sentence.

In the process of exploring the high value-added utilization of FA, the chemical composition of fly ash (wt%SiO2+wt%Al2O3>70%) was found to be extremely similar to that of zeolite.

Piont 9: Page 2, line 73 define secondary pollution.

Response 9: Thank you for your careful review. What we want to express is that NASH, which produces more impurities in industrial production, is a pollutant.

Piont 10: Page 2, Line 75, remove “for resource utilization” because it was in the previous line

Response 10: Thank you for your advice. We have modified in the text.

Piont 11: Page 2, Line 78, “experiences”

Response 11: Thank you for your advice. We have modified in the text.

Point 12: Page 2, Line 81, “post-modification”

Response 12: Thank you for your advice. We have modified in the text.

Piont 13: Page 2, Line 83 change "shows an exploration" to examines

Response 13: Thank you for your advice. We have modified in the text

Piont 14: Page 2, Line 85, “analyses of the adsorbed precipitates”

Response 14: Thank you for your advice. We have modified in the text.

Piont 15: Page 2, Line 91 define AR

Response 15: Thank you for your advice. We have defined in the text. Analytically reagent (AR ≥99.5% purity).

Piont 16: Page 3, Line 99, what constant weight? What’s the starting mass of FA?

Response 16: Raw fly ash may contain moisture due to exposure to the air, so we need to dry it. Here "constant weight" means to dry until the sample does not contain moisture. Generally, 100g of raw ash was selected for drying, and the mass was generally reduced by 0.01-0.1%, which is the water content.

Piont 17: Page 3, Line 100, change “function” to “purpose”

Response 17: Thanks for your suggestion, we have changed "function" to "purpose" in the article.

Piont 18: Page 3, line 105: This sentence needs to be reworded “stirred at an even speed of 80°C”. I am interested to know at what speed this was performed.

Response 18: The liquid-solid mixture was stirred at a uniform speed of 120-140 rpm/min at 80°C for 6 hours.

Piont 19: Page 3, line 113: should be XRPD X-ray powder diffraction.

Response 19: Thank you for your advice. We have modified in the text.

Piont 20: Page 3, line 130: What was the constant temperature used during stirring for the batch experiment? What are AR and GR?

Response 20: The stirring was carried out at 25°C and room temperature. Analytically reagent (AR ≥99.5% purity), guaranteed reagent (GR ≥99.9% purity), already defined in the text.

Piont 21: Page 3, line 132. Please define Cp and Cn. Also, define any other abbreviations in the text.

Response 21: Thank you for your advice. We have paid attention to defining all abbreviations.

Piont 22: Page 3, Line 133, rated concentration? Written? what do the authors mean by preconfigured? Was a pH meter used?

Response 22: The meaning of the rated concentration is the concentration required for the experiment, saying that we have not stated clearly that it has been corrected. In order to facilitate the experiment, we configured a higher concentration stock solution, this process did not use a pH meter.

Piont 23: Page 3, line 142. Explain why the initial ammonium and phosphate concentrations are configured to be 50 mg N/L and 10 mg P/L. Only the reference is not enough. Explain that N represents ammonium and P phosphate.

Response 23: In order to meet the actual wastewater treatment, we selected the initial ammonium and phosphorus concentrations through investigation. This study also continued to study the adsorption conditions at different concentrations. We have defined'"N" and "P".

Piont 24: Line 145 – Page 4, Line 148, sentences are not clear

Response 14: Thank you for your advice. We have modified in the text.

Piont 25: Page 4, line 160-164 is Qe unitless? How does Ce relate to Ct? How do the units cancel when C is in mg/L and W in g?

Response 25: The unit of Qe is mg/g. C0 is the concentration of solute in the initial solution, and Ce is the concentration of solute in the solution after adsorption. Because of the adsorption reaction, the C0-Ce can know how much solute has been adsorbed, and there is a variable relationship between the two. The unit of C is mg/L, the unit of V is L, C×V can know how many solutes in the solution are in mg. C×V/W can get the amount of solute adsorbed per gram of adsorbent, that is, the unit adsorption capacity Qe, the unit is mg/g.

Piont 26: Page 4, line 169 results show

Response 26: Thank you for your advice. We have modified in the text.

Piont 27: Page 4, line 171 & 174 should be broad not bread peak

Response 27: Thank you for your advice. We have modified in the text.

Piont 28: Page 4 equations need states of matter such as

Response 28: Thanks for your suggestion, we have added the state of matter to the equation

Piont 29: Page 5, line 177: add more description to the caption. Add the space between intensity and AU- correct this in all y-axes of the figures.

Response 29: Thank you for your advice. We have modified in the text, and added more description to the title of the figure

Piont 30: Page 5, line 179 Please articulate and better explain the sentence: The chemical composition of the synthesized product is basically similar to that of 179 the original ash (Table 1), and the molecular ratio of the formed sodium aluminosilicate 180 solution is on the high side.

Response 30: The meaning here is the formation of sodium aluminosilicate with a lower silicon to aluminum ratio (SiO2/Al2O3). We have modified in the text.

Piont 31: Page 6, line 205. ‘’The morphological analysis of fly ash’’ is not properly correct since the authors are only showing images and not performing any types of analysis. Comment on the surface area changes from SEM images.

Response 31: Thank you for your advice. We have modified in the text.

Piont 32: Page 6 line 210: Define and explain BET. Authors claim they examined the internal pore structure and specific surface area of the materials by N2 adsorption BET by use of a Micro-meritics ASAP 2020 volumetric instrument in the methods. Add more details.

Response 32: Thank you for your advice. We reorganized the content and added specific results of the BET test in terms of characterizing materials in the text. In the article, we added N2 adsorption/desorption isotherms and pore size distribution maps, and compared the raw materials and the synthesized products.

Piont 33: Page 6, Line 204, Figure 2 needs a few sentences of description (caption). Also needs to be bigger because it is difficult to read. The legend says MFA and FA, but the caption states FA and SOD.

Response 33: Thank you for your advice. We have corrected the labeling error in the article and rewritten the title.

Piont 34: Page 6, Line 209, Figure 3 needs a few sentences of description (caption). Also should be A, B, C, D. Don't use blue on a black background.

Response 34: Thank you for your suggestion, we have corrected it in the picture.

Piont 35: Page 6, Line 212, define CEC

Response 35: Thank you for your correction, we have defined the cation exchange capacity as CEC.

Piont 36: Page 7, lines 227. Why is b discussed before a? Correct this. How authors calculated the linear relationship between the unit adsorption capacity and the removal rate with dose should be added here.

Response 36: Thank you for your correction. Sorry for not discussing in order, we have corrected it. As the amount of adsorbent increases, the removal rate increases, and the removal rate can be calculated from equation (2). According to the removal rate, the actual adsorbed solute content can be calculated. From the equation (1), the ratio of the adsorbed content to the dose is the unit adsorption capacity.

Piont 37: Page 7, Line 228-229, this is not a complete sentence. There is no need to state again “from the figure” after the sentence before it ended with “can be seen in the figure”

Response 37: Thank you for your advice. We have rewritten the sentence to correct it.

Piont 38: Page 7 line 232-234: This sentence is confusing. You state that when the dosage is lower than 3 g·L the removal rates increase as the dosage increases. Do you mean that 3 g·L is the max does to see an increase uptake of ammonium and phosphate or is that when first start to see an uptake?

Response 38: In Figure 4a, the larger the slope of the curve, the faster the adsorption rate. When the adsorbent dosage is 0-3g/L, the adsorption rate is faster because there are more vacancies in the adsorption point. Continue to increase the dosage and find that the removal rate does not increase significantly, and the unit adsorption capacity decreases, indicating that there is already adsorption saturation.Continue to increase the dosage will result in waste of resources, so considering the better removal effect and environmental protection, the adsorbent dosage is 4g/L.

Piont 39: Page 7, Line 235, missing L-1 in 6g*L

Response 39: Thank you for your careful review. We have made corrections in the text.

Piont 40: Page 7, Line 237, “decreases”

Response 40: Thank you for your advice. We have modified in the text.

Piont 41: Page 7, line 250. Please define TP TN. Also I wonder if there is a better way to graphically represent the 2 graphs since they are plotting on the same x-axis.

Response 41: Regarding the usage and definition of abbreviations, I have corrected them in the text. As Figure a and Figure b respectively represent the removal rate and unit adsorption capacity, in order to explore the change law more clearly, the two are drawn separately.

Point 42: Figure 4: the legend improvements. Please specify what is TN and TP. What is the error?

Response 42: Thank you for your advice. We have modified in the text. The error of TN and TP is due to the three sets of parallel experiments we conducted.

Piont 43: Page 8 line 269: please specify what is “the material” is this SOD?

Response 43: Thank you for your advice. We have modified in the text, defined material as SOD.

Piont 44: Page 10, Line 310, Figure 7 needs a few sentences of description (caption). What is the blue arrow pointing to and why is this important? A scale bar would have been nice.

Response 44: Thank you for your advice. We have modified in the text. Thank you for your advice. For Figure 7, we mainly want to express the precipitation and flocs formed when adsorbing high concentrations of phosphate. As can be seen in the figure, the gray matter is glued together.

In the above figure, the beaker on the left is the SOD in deionized water, and the beaker on the right is the SOD after phosphate adsorption and settling. It can be seen that obvious flocs are formed after adsorption, and there are sediments in the flocculation.

Piont 45: Page 10, Line 310, Figure 8 needs a few sentences of description (caption). Difficult to read.

Response 45: Thank you for your advice. We have defined in the text.

Piont 46: Page 10, Line 332, ICP is not mentioned in methods section. OES or MS as detector? Also needs to be defined.

Response 46: Thank you for your advice. We have added ICP-OES test content in the method section.

Piont 47: Page 11, line 349-350 subscript solution is misspelled. Assign states of matter to the equation.

Response 47: Thanks for your suggestion, we corrected this equation and added the state of matter.

Piont 48: Page 11 line 333: Which ICP is used OES or MS? Clarify.

Response 48: Thanks for your suggestion, we were used ICP-OES which has been modified in the text.

Piont 49: Page 14, Line 422, Figure 11 needs a few sentences of description (caption).

Response 49: Thank you for your advice. We have modified in the text.

Piont 50: Page 15, Line 310, Figure 7 needs a few sentences of description (caption).

Response 50: Thank you for your advice. We have modified in the text.

Piont 51: Page 16, Line 472, why suddenly change from Celsius to Kelvin?

Response 51: Celsius is changed to Kelvin because the unit needs to be changed to "K" for the calculation of thermodynamic parameters.

Piont 52: Page 16, Line 487, cmol?

Response 52:Thank you for your advice. Here "cmol/kg" is the unit of cation exchange capacity, and it can also be changed to "mmol/kg".

Piont 53: Page 16, Line 489, not a complete sentence

Response 53: Thank you for your correction, we have added the subject in the text.

Piont 54: Page 16, Line 494, not a complete sentence

Response 54: Thank you for your advice. We have modified in the text.

Piont 55: Page 16, Line 498, not a complete sentence

Response 55: Thank you for your advice. We have modified in the text.

Piont 56: Page 16 section 4: Use of bullet points for the conclusion is not appropriate. Change this to a paragraph format.

Response56: We have changed the bullets in the conclusion section to paragraph format.

Piont 57: Another general comment is to add a section indicating how you calculate the fitting models (software used, methods etc)

Response 57: Thank you for your advice. In order to facilitate reading, we choose to combine model calculations with the main text.

Piont 58: Page 18, Line 605-606, citation format not consistent

Response 58: Thanks for your suggestion, we have modified the format of references.

Piont 59: Page 18, Line 607-608, citation format not consistent

Response 59: Thank you for your advice. we have modified the format of references.

Round 2

Reviewer 2 Report

The authors have made the suggested corrections. The manuscript is now of sufficient quality to be published in materials.

Author Response

Point: The authors have made the suggested corrections. The manuscript is now of sufficient quality to be published in materials.

Response: The authors would like to thank Reviewer 1 for comprehensive criticism and suggestions. We will continue to work hard to revise the article to improve the quality of the article.

Reviewer 5 Report

The authors do not know what should be in past tense and what should be in current tense. The manuscript still needs extensive english editing before additional review.

Page 1 line 27 excellent good are both written. Remove one of them

Page 1 line 32 should be bodies "is" and "are" present in the water. 

Page 1 line 36 Should be "and if the ammonium ion..exceed the standard...this will lead to"

Page 2 line 52 what are the authors trying to say? what is a superimposed influence?

Page 2 Line 70 should be "to modify the sediments"

Page 3 line 102,  characterized by what methods? What common absorption models?

Page 4, line 180 is not a complete sentence and does not make sense

Page 5, line 202 Include the unit for Qe (mg/g)

Page 5 line 207 should be "A standard stock solution of xyz was diluted.." What was the standard stock solution? "4 g/L of SOD was added to each of the two solutions...." What are the "best absorption conditions in the experiment"?

Page 5 equation  3 add a space between Desorption and rate

Page 6 line 231 need to be clear that both Qr  and  Qmax  are in units of mg/g

Page 6 line 235  should be "are shown in Figure 1"

Page 6 line 240 should be heating

Page 6 line 242 should be broad

Page 6 equations 5 and 6 all ions need to have (aq)

Page 6 what are each of the bragg peaks (ie 111, 100, 110? that correspond in the figure?) identifying the component is not sufficient. Maybe add a SI table for this.

Page 7 figure 2 Add a space between transmittance and % on the y-axis. Clarify what T-O represents is in the figure caption.

Page 8-9 Figure 3 each images needs a different letter identifier in the top corner. In the caption x1000 times is redundant. Remove the word microscopic  and microscopic view from the caption as SEM stands for microscopy.

Page 9 Line 282  first define BET and what it is used  to measure, then comment on the data. First  define BJH and what it is used  to measure, then comment on the data.

page 10 line 332, Page 11 line 336, and Page 12  line 374  Change should be capitalized.

page 12 line 380 "analysis of the causes" does not make sense.

page 12 Line 382 making was the correct tense and resulting as  well

Page 12 Equations 7-10 are  missing (aq) as states of matter for each ion. Why are = signs used and not equilibrium arrows?

Page 13 line 339 use the term flocculation not flocs in the caption

Page  13 figure 9 correct the y axis with a space after intensity and before (A.U.) Label the bragg peaks or add a SI table.

Figure 10 add a space between removal and %. State in the figure caption what the error bars are from,  ie. SEM of three independent samples?

Figure  13 what  are  the units for the axes?

Figure  14 add a space after the word percentage and before (%) same for the legend items

page 20 Correct the spacing in the conclusions between the paragraphs

Page 20 line 603 should be "to investigate"... in  wastewater, we modified adsorbent dosage... Finish this first sentence or combine with the second. As written it does not make sense.

Page 21  line 626 a paragraph should have at least 3-5 sentences.

Author Response

The authors do not know what should be in past tense and what should be in current tense. The manuscript still needs extensive english editing before additional review.

The authors would like to thank Reviewer 5 for the critical and constructive comments. We have made careful revisions based on the suggestions, and hope to get your approval. We have revised the English grammar again.

Point 1: Page 1 line 27 excellent good are both written. Remove one of them

Response 1: Thank you for correcting the repetitive statement. We have revised it in the article.

Point 2:Page 1 line 32 should be bodies "is" and "are" present in the water. 

Response 2: Thank you very much for your suggestion, we have made changes in the text.

Point 3: Page 1 line 36 Should be "and if the ammonium ion..exceed the standard...this will lead to"

Response 3: Thank you very much for your suggestion, we have made changes in the text.

Point 4: Page 2 line 52 what are the authors trying to say? what is a superimposed influence?

Response 4: Thanks for the questions raised by the reviewers, we will give some explanations here. Fly ash is the most common solid waste generated from coal-fired power generation. Under normal circumstances, fly ash is roughly composed of aluminum oxide, silicon oxide and iron oxide. Because of the different factors such as the origin of the coal mine, the type of coal, the type of furnace used for burning the coal, and the way of discharging ash, the chemical composition of various fly ash will be different. Moreover, the reasons for these formations are different and combined with each other, which will have a superimposed influence on the formation of fly ash. Fly ash produced in the same batch may become very different in composition due to different combustion temperatures or other reasons. The combination of many factors makes fly ash complex and contains toxic elements, so the large-scale utilization rate is generally low. We have already wanted to show in the article that there are many reasons that affect the composition of fly ash, and it is toxic to the environment leading to low utilization.

Point 5: Page 2 Line 70 should be "to modify the sediments"

Response 5: Thank you very much for your suggestion, we have made changes in the text.

Point 6: Page 3 line 102,  characterized by what methods? What common absorption models?

Response 6: Thanks to the reviewers for their suggestions. In the experiments, the original and modified samples were systematically characterized by XRD, XRF, FTIR, SEM, etc. Common adsorption models such as Lagergren first-order, Ho'pseudo-second-order, Langmuir isotherm model, Freundlich isotherm model, etc. were selected to describe the adsorption process.

Point 7: Page 4, line 180 is not a complete sentence and does not make sense

Response 7: Thank you for your suggestion, we have corrected it in the article.

Point 8: Page 5, line 202 Include the unit for Qe (mg/g)

Response 8: Thanks for your suggestion, we have added the Qe unit.

Point 9: Page 5 line 207 should be "A standard stock solution of xyz was diluted.."  What was the standard stock solution? "4 g/L of SOD was added to each of the two solutions...." What are the "best absorption conditions in the experiment"?

Response 9: Thank you for your suggestion, we have corrected it.

A standard stock solution of ammonium ion and phosphate (CN=1000mg/L, CP=500mg/L) were diluted with deionized water. Dilute to the concentration of ammonium ion and phosphate respectively CN=500mg/L and CP=200mg/L.

The best adsorption conditions are obtained from the preliminary batch adsorption experiments. In order to explore the desorption under saturated adsorption, the adsorption conditions at T=25℃ and pH=8 are selected.

Point 10: Page 5 equation  3 add a space between Desorption and rate

Response 10: Thanks for your suggestion, we have modified it in the formula.

Point 11: Page 6 line 231 need to be clear that both Qr  and  Qmax  are in units of mg/g

Response 11: Thank you for pointing out our shortcomings. We have marked the units of each parameter in the text.

Point 12: Page 6 line 235  should be "are shown in Figure 1"

Response 12: Thank you for your suggestion, we have corrected it.

Point 13: Page 6 line 240 should be heating

Response 13: Thank you for your correction, we have rewritten this sentence.

Point 14: Page 6 line 242 should be broad.

Response 14: Thank you for your suggestion, we have corrected it.

Point 15: Page 6 equations 5 and 6 all ions need to have (aq)

Response 15: Thank you for your suggestion. We have corrected Equation 5 and Equation 6.

Point 16: Page 6 what are each of the bragg peaks (ie 111, 100, 110? that correspond in the figure?) identifying the component is not sufficient. Maybe add a SI table for this.

Response 16: Thanks to the reviewer’s suggestions, we have added the crystal planes in the picture and added the SI table.

Point 17: Page 7 figure 2 Add a space between transmittance and % on the y-axis. Clarify what T-O represents is in the figure caption.

Response 17: Thank you for your suggestion, we have modified the figure and title.

Point 18: Page 8-9 Figure 3 each images needs a different letter identifier in the top corner. In the caption x1000 times is redundant. Remove the word microscopic  and microscopic view from the caption as SEM stands for microscopy.

Response 18: Thank you for your suggestion, we have modified the figure and title.

Point 19: Page 9 Line 282  first define BET and what it is used  to measure, then comment on the data. First  define BJH and what it is used  to measure, then comment on the data.

Response 19: Thank you for your suggestion, we have made changes in the text.

Point 20: page 10 line 332, Page 11 line 336, and Page 12  line 374  Change should be capitalized.

Response 20: Thanks for your suggestion, we have modified the title of the picture.

Point 21: page 12 line 380 "analysis of the causes" does not make sense.

Response 21: Thank you very much for pointing out the error, we have already corrected it in the article.

Point 22: page 12 Line 382 making was the correct tense and resulting as well

Response 22: Thank you for your suggestion, we have made changes in the article.

Point 23: Page 12 Equations 7-10 are  missing (aq) as states of matter for each ion. Why are = signs used and not equilibrium arrows?

Response 23: Thank you for your suggestion, we have made changes in the article. We have modified the symbol because these reactions are reversible.

Point 24: Page 13 line 339(366) use the term flocculation not flocs in the caption

Response 24: Thanks for your suggestion, we have changed "flocs" to "flocculation" in the title.

Point 25: Page  13 figure 9 correct the y axis with a space after intensity and before (A.U.) Label the bragg peaks or add a SI table.

Response 25: Thank you for your suggestion, we have added the SI table.

Point 26: Figure 10 add a space between removal and %. State in the figure caption what the error bars are from,  ie. SEM of three independent samples?

Response 26: Thanks for your suggestion, we have modified the figure and explained the source of the error bars.

Point 27: Figure  13 what  are  the units for the axes?

Response 27: The units of Ce and Qe are mg/g. However, due to the logarithmic operation performed by the isothermal model, the unit does not need to be considered after the logarithmic conversion, but the numerical value is calculated.

Point 28: Figure 14 add a space after the word percentage and before (%) same for the legend items

Response 28: Thank you for your suggestion, we have modified it in the picture.

Point 29: page 20 Correct the spacing in the conclusions between the paragraphs

Response 29: Thanks for your suggestion, we have checked the paragraph layout of the full text.

Point 30: Page 20 line 603 should be "to investigate"... in  wastewater, we modified adsorbent dosage... Finish this first sentence or combine with the second. As written it does not make sense.

Response 30: Thank you for your suggestion, we have revised the sentences in the paragraph.

Point 31: Page 21  line 626 a paragraph should have at least 3-5 sentences.

Response 31: Thanks for your suggestion, we have adjusted the sentences in the paragraph.
